# Psychometric properties of measures of upper limb activity performance in adults with and without spasticity undergoing neurorehabilitation–A systematic review

**Shannon Pike** [1,2], **Anne Cusick**[3], **Kylie Wales** [4], **Lisa Cameron**[5], **Lynne Turner-Stokes** [6,7], **Stephen Ashford**[6,7,8], **Natasha A. Lannin**[1,5,9] *

1 School of Allied Health, Human Services and Sport (Occupational Therapy), La Trobe University, Melbourne, Victoria, Australia, 2 Wagga Wagga Ambulatory Rehabilitation Service, Murrumbidgee Local Health District, Wagga Wagga, New South Wales, Australia, 3 Discipline of Occupational Therapy, Faculty of Medicine and Health, The University of Sydney, Sydney, New South Wales, Australia, 4 School of Health Sciences, Faculty of Health and Medicine, The University of Newcastle, Newcastle, New South Wales, Australia, 5 Alfred Health, Melbourne, Victoria, Australia, 6 Regional Hyper-acute Rehabilitation Unit, London North West University Healthcare NHS Trust, Northwick Park Hospital, London, United Kingdom, 7 King's College London, Department of Palliative Care, Policy and Rehabilitation, London, United Kingdom, 8 Centre for Nursing, Midwifery and Allied health led Research, University College London Hospitals, National Hospital for Neurology and Neurosurgery, London, United Kingdom, 9 Department of Neurosciences, Central Clinical School, Monash University, Melbourne, Victoria, Australia

* Natasha.Lannin@monash.edu

**Data Availability Statement:** All relevant data are within the manuscript and its Supporting Information files.

## Abstract

### Introduction

This systematic review appraises the measurement quality of tools which assess activity and/or participation in adults with upper limb spasticity arising from neurological impairment, including methodological quality of the psychometric studies. Differences in the measurement quality of the tools for adults with a neurological impairment, but without upper limb spasticity, is also presented.

### Methods

29 measurement tools identified in a published review were appraised in this systematic review. For each identified tool, we searched 3 databases (Medline, Embase, CINAHL) to identify psychometric studies completed with neurorehabilitation samples. Methodological quality of instrument evaluations was assessed with use of the Consensus-based Standards for the Selection of Health Status Measurement Instruments (COSMIN) checklist. Synthesis of ratings allowed an overall rating of the psychometric evidence for each measurement tool to be calculated.

### Results

149 articles describing the development or evaluation of psychometric properties of 22 activity and/or participation measurement tools were included. Evidence specific to tool use for

**Funding:** This work was supported by an Australian Government Research Training Program Scholarship (SP); NAL was supported by a Future Leader Fellowship (102055) from the National Heart Foundation of Australia. The funders had no role in study design, data collection and analysis, decision to publish, or preparation of the mauscript.

**Competing interests:** The authors have declared that no competing interests exist.

adults with spasticity was identified within only 15 of the 149 articles and provided evidence for 9 measurement tools only. Overall, COSMIN appraisal highlighted a lack of evidence of measurement quality. Synthesis of ratings demonstrated all measures had psychometric weaknesses or gaps in evidence (particularly for use of tools with adults with spasticity).

## Conclusions

The systematic search, appraisal and synthesis revealed that currently there is insufficient measurement quality evidence to recommend one tool over another. Notwithstanding this conclusion, newer tools specifically designed for use with people with neurological conditions who have upper limb spasticity, have emergent measurement properties that warrant further research.

## Systematic review registration

PROSPERO CRD42014013190.

## Introduction

The personal experience of a neurological condition can be profound, impacting on all areas of a person's health and wellbeing. The International Classification for Functioning Disability and Health (ICF) [1] provides a framework to consider the impact of a neurological condition on a person, highlighting both the breadth and complexity of potential issues. While the ICF can classify areas that may be impacted by neurological conditions, and some rating of impairment and limitation is possible using the ICF core sets [2, 3], precise measurement of factors known to be related to activity is essential.

Measurement is key to determining the effect of rehabilitation interventions, and therefore measurement tools used in neurorehabilitation should target all levels of functioning, disability and health–this includes activity and participation as much as impairments in body structure and function [4]. In addition to targeting all levels, measurement should also capture and reflect actual performance of everyday 'real-life' activities outside of the clinical setting [5]. Measurement of activity and participation in 'real-life' activities presents many challenges, not least of which is consistency, validity and sensitivity of 'real life' functions.

Several reviews have sought to identify and determine the most suitable measures to evaluate upper limb impairment and activity for adults with a neurological condition [5–7]. Scant evidence has been located and clear gaps have been identified in the presentation of the psychometric quality of the tools in a neurorehabilitation context. Furthermore, Alt Murphy [6], identified many of the included reviews failed to critically appraise the methodological quality of the individual studies evaluating the psychometric properties of the tools. Whilst recommendations regarding upper limb evaluation have been made, the tools identified and the evidence regarding the psychometric properties of the tools were not specifically targeted nor extracted from a sample of adults with upper limb spasticity as a result of their neurological condition.

Review work by members of this study's authorship team, Ashford and Turner-Stokes, did identify outcome measurement tools both applicable to the upper limb that assess function in the context of everyday life, and from studies including adults with upper limb spasticity [8]. They demonstrated newer upper limb measurement tools used in neurorehabilitation research

which examine activity and participation in the context of everyday real-life activities show promise [8]. There is thus a need for a comprehensive appraisal and synthesis of the psychometric properties of all these tools, to potentially recommend a tool/s for clinical and research use.

The two aims of this study, therefore, was to firstly critically appraise and summarize the quality of the psychometric properties of previously identified upper limb activity performance measurement tools [8] when used with adults with upper limb spasticity using a level of evidence approach and the COnsensus-based Standards for the selection of health Measurement INstruments (COSMIN) guidelines [9–11]. Secondly, to determine if the presence of upper limb spasticity impacts on which measure should be selected based on psychometric evidence, differences in psychometric properties for the identified measurement tools for adults with a neurological impairment but without upper limb spasticity will be defined.

## Method

A systematic review with COSMIN appraisal was undertaken, with PRISMA guidelines informing reporting.

### Identification and selection of measurement tools

The published list of measurement tools by Ashford and Turner-Stokes [8] was used to identify and select measurement tools for appraisal. The effect of upper limb spasticity on gait is acknowledged [12]. However, we delimit this review to measurement tools that assess upper limb functional movement. As this source systematic review was published in 2013, the most recent clinical guidelines management of spasticity in the upper limb [13] was also searched so as to identify any potential tools that assess upper limb functional movement which may have been developed since 2013. One further tool, the Arm Activity Measure (ArmA), was located and subsequently included in the review.

### Measurement tool inclusion criteria

To be included, measurement tools had to assess activity or performance as defined by the ICF [1], and each needed to focus on the upper limb. Activity is defined within the ICF as "the execution of a task or action by an individual" [1, p10] while participation is defined as "involvement in a life situation" [1, p10]. In the present study, the official World Health Organisation (WHO) coding of activity and participation was used, that of a single overlapping list of categories [14]; tools that only evaluate impairment/s (e.g. pain, range of movement, contracture, spasticity) were excluded.

### Study search strategy

Searches were completed per protocol [15] to identify research that administered the measurement tool with adults who had neurological conditions. The search was run in Medical Literature Analysis and Retrieval System Online (MEDLINE), Cumulative Index to Nursing and Allied Health Literature (CINAHL) and Excerpta Medical database (EMBASE) from inception to December 2016. Where able, the validated search filter for finding studies on measurement properties was used [16]; search terms are presented in S1 File. COSMIN requires information regarding the development/content validity of the measurement tools to be sought, therefore tool references were identified and obtained when not identified within the search results.

## Study screening

Title and abstracts were downloaded into the reference management system EndNote™. Duplicates were removed and screened for inclusion by one reviewer. To minimize the risk of incorrect inclusion and exclusion of studies; a second reviewer screened a random 25% sample of included studies against inclusion criteria and all excluded papers were reviewed by the senior author. Disagreements were settled through independent review, followed by discussion until a consensus decision was reached. Full text papers were obtained for all included studies and checked to confirm the final inclusion/exclusion decision [15].

## Study inclusion and exclusion criteria

Studies which included participants both with and without spasticity were included; to be included in the spasticity analysis, evidence of the presence of participant upper limb spasticity was required—not just the mention of 'spasticity' in text. For example, the study by Page, Levine and Hade [17] reported a Modified Ashworth Scale score of $\geq 3$ as an exclusion criterion; but within the study sample there was no evidence of participants with spasticity $\leq 3$. Thus, this article was deemed to be a study without upper limb spasticity. In addition, only studies which tested the measurement tool in its *original and complete* form were included. This conservative approach to study selection was taken to ensure maximum possible homogeneity in the evidence base which would be used to underpin tool recommendations for practice use. If a tool was used as a comparator to validate another tool, the study was excluded in accordance with COSMIN methodology. *Full protocol has been published elsewhere.* Inclusion criteria are detailed in Table 1.

## Data analysis

**Methodological quality of studies.** The quality of the included studies was appraised using the COSMIN taxonomy of measurement properties and definitions for health-related patient reported outcomes [9–11] and the COSMIN Risk of Bias checklist [18] for systematic reviews of patient-reported outcome measures. The methodological quality of each study was individually assessed to evaluate whether it met the standards for measurement tool development, content validity, structural validity, internal consistency, cross-cultural validity/measurement invariance, reliability, measurement error, criterion validity, hypothesis testing for

**Table 1. Inclusion criteria.**

| Design |
| --- |
| • Psychometric properties of the identified measurement tools were evaluated |
| • Original research |
| • Conducted and published in English within peer reviewed literature |
| **Participants** |
| • Adults (>18 years old) |
| • ≥ 90% diagnosis of a following neurological condition; Stroke, Multiple Sclerosis, Cerebral Palsy, Traumatic Brain Injury, Anoxia |
| • With or without upper limb spasticity |
| • Undergoing rehabilitation |
| **Measurement tool** |
| • Measured activity and/or participation |
| • Nil modifications |
| • Complete measure administered |

construct validity and responsiveness. The Risk of Bias checklist rated each measurement property as either "very good", "adequate", "doubtful" or "inadequate". As there is no accepted "gold standard" measure of upper limb activity, criterion validity was not evaluated, and construct validity and responsiveness properties were appraised within the hypothesis testing criteria of COSMIN. Where *a priori* hypotheses were not stated, studies were assigned an appropriate generic hypothesis from the list developed by the COSMIN group [18]. Information regarding the interpretability and generalizability were collected.

**Quality of measurement properties.**   The results of individual studies reporting on the psychometric properties were then evaluated using Terwee's quality criteria for measurement properties [9], see S1 File. Results were rated as sufficient '+', indeterminant '?'or insufficient '-'.

**Sample size of studies.**   Sample size was only assessed within individual studies evaluating the measurement properties of content validity, structural validity and cross-cultural validity as per COSMIN guidelines. Sample sizes of individual studies evaluating the remaining measurement properties were not assessed via the Risk of Bias Checklist, and sample sizes per those measurement properties were instead pooled at the synthesis stage [9].

**Synthesis of best evidence.**   All identified evidence and results were then pooled and the modified COSMIN GRADE approach used to determine the overall quality of the evidence [9]. The modified COSMIN GRADE approach considers and downgrades the level of evidence and consequently trustworthiness of results depending on the risk of bias (methodological quality), inconsistency of results, imprecision (based on total sample size) and indirectness (evidence from different populations than the population of interest) [9, p1151]; indirectness was not applicable in this review as studies conducted in samples other than those specified in the inclusion and exclusion criteria were excluded. The synthesis determines either "high", "moderate" "low" or "very low" quality levels of 'sufficient', 'insufficient', 'inconsistent' or 'indeterminant'.

## Results

Of the 33 measurement tools identified in the Ashford and Turner-Stokes review [8], 29 measurement tools were published tools. One of the published tools, the Ten Metre Walk Test, was excluded as it does not directly assess upper limb functional movement or use. We therefore completed searches for these 28 tools plus the ArmA (which was identified in the clinical guideline review), resulting in 29 tools in total.

### Flow of studies

The electronic search strategy located 55,679 studies across the individual measurement tools. After screening titles, abstracts and full text, 149 psychometric studies (some evaluating more than one included tool) were included in this systematic review. Our systematic search did not locate any studies evaluating the psychometric properties of the following: Frenchay Arm Test [19], Global Assessment Scale [20], Goal Attainment Scale– 10 point scale [21], Klein-Bell Activities of Daily Living Scale [22], Motor Activity Log-5 [23], Leeds Adult Spasticity Impact Scale [24] and Patient Disability Scale/Carer Burden Scale [24]. Fig 1 presents the flow of papers through the review.

### Characteristics of the studies

The 149 included studies are outlined in Table 2. The majority of studies (n = 91, 61%) included post-stroke participants, and of these, most were greater than 6 months post-stroke. The remaining studies included diagnoses of multiple sclerosis (MS), traumatic brain injury (TBI) or mixed neurological participants. Sample characteristics varied across studies and

**Identification**

Records identified through database searching

CINAHL
(ARAT n=69, ArmA n=26, AQoL n=122, BI n=1129, CMSA n=25, DAS n=123, EQ-5D n=456, FAT n=1, mFAT n=18, FIM n=840, GAS n=96, GAS–10pt n=0, Global Ax n=157, KleinBell ADL n=8, LASIS n=4, SF-36 n=3125, MAL n=60, MAL-5 n=0, MAL-28 n=1, MI n=46, NHPT n=53, OHS n=188, PDS/CBS n=0, RMA n=34, RMA-UL n=34, SA-SIP n=9, SIS n=69, UL MAS n=76)

EMBASE
(ARAT n=344, ArmA n=99, AQoL n=920, BI n=3492, CMSA n=70, DAS n=111, EQ-5D n=4175, FAT n=5, mFAT n=38, FIM n=2088, GAS n=313, GAS–10pt n=0, Global Ax n=442, KleinBell ADL n=7, LASIS n=3, SF-36 n=14374, MAL n=220, MAL-5 n=0, MAL-28 n=10, MI n=127, NHPT n=628, OHS n=49, PDS/CBS n=0, RMA n=230, RMA-UL n=230, SA-SIP n=15, SIS n=256, UL MAS n=37)

MEDLINE
(ARAT n=297, ArmA n=150, AQoL n=589, BI n=1619, CMSA n=43, DAS n=84, EQ-5D n=5150, FAT n=4, mFAT n=26, FIM n=1766, GAS n=218, GAS–10pt n=0, Global Ax n=427, KleinBell ADL n=7, LASIS n=1, SF-36 n=8942, MAL n=184, MAL-5 n=0, MAL-28 n=3, MI n=146, NHPT n=234, OHS n=25, PDS/CBS n=0, RMA n=265, RMA-UL n=265, SA-SIP n=13, SIS n=137, UL MAS n=20)

Records identified through other sources/measures searches
(ARAT n=1, ArmA n=0, AQoL n=0, BI n=1, CMSA n=2, DAS n=0, EQ-5D n=1, FAT n=0, mFAT n=0, FIM n=3, GAS n=0, GAS–10pt n=0, Global Ax n=0, KleinBell ADL n=0, LASIS n=0, SF-36 n=3, MAL n=1, MAL-5 n=0, MAL-28 n=0, MI n=1, NHPT n=1, OHS n=0, PDS/CBS n=0, RMA n=1, RMA-UL n=1, SA-SIP n=1, SIS n=0, UL MAS n=1)

**Screening**

Records screened after duplicates removed
(ARAT n=412, ArmA n=214, AQoL n=1261, BI n=4866, CMSA n=82, DAS n=263, EQ-5D n=7064, FAT n=6, mFAT n=60, FIM n=3309, GAS n=409, GAS–10pt n=0, Global Ax n=1026, KleinBell ADL n=15, LASIS n=4, SF-36 n=16507, MAL n=293, MAL-5 n=0, MAL-28 n=10, MI n=204, NHPT n=585, OHS n=262, PDS/CBS n=0, RMA n=389, RMA-UL n=389, SA-SIP n=22, SIS n=301, UL MAS n=102)

Records excluded at abstract
(ARAT n=357, ArmA n=207, AQoL n=1257, BI n=4734, CMSA n=57, DAS n=250, EQ-5D n=7025, FAT n=4, mFAT n=39, FIM n=3164, GAS n=360, GAS–10pt n=0 , Global Ax n=707, KleinBell ADL n=9, LASIS n=2, SF-36 n=16457, MAL n=257, MAL-5 n=0, MAL-28 n=7, MI n=168, NHPT n=497, OHS n=200 , PDS/CBS n=0, RMA n=358, RMA-UL n=358, SA-SIP n=15, SIS n=235, UL MAS n=80)

**Eligibility**

Full-text articles assessed for eligibility
(ARAT n=55, ArmA n=7, AQoL n=5, BI n=132, CMSA n=25, DAS n=13, EQ-5D n=39, FAT n=2, mFAT n=21, FIM n=145, GAS n=49, GAS–10pt n=0, Global Ax n=6, KleinBell ADL n=6, LASIS n=2, SF-36 n=53, MAL n=36, MAL-5 n=0, MAL-28 n=3, MI n=36, NHPT n=88, OHS n=24, PDS/CBS n=0, RMA n=31, RMA-UL n=31, SA-SIP n=7, SIS n=66, UL MAS n=22)

Full-text articles excluded*
(ARAT n=43, ArmA n=2, AQoL n=2, BI n=117, CMSA n=21, DAS n=11, EQ-5D n=20, FAT n=2, mFAT n=19 , FIM n=125, GAS n=40, GAS–10pt n=0, Global Ax n=6, KleinBell ADL n=6, LASIS n=2, SF-36 n=29, MAL n=32, MAL-5 n=0, MAL-28 n=2, MI n=30, NHPT n=80, OHS n=23 , PDS/CBS n=0 , RMA n=26, RMA-UL n=25, SA-SIP n=3, SIS n=56, UL MAS n=12)

**Included**

Studies included in quantitative synthesis
(ARAT n=12, ArmA n=5, AQoL n=3, BI n=6, BI n=6, BI(C&W) n=9, CMSA n=4, DAS n=2, EQ-5D n=19, FAT n=0, mFAT n=2, FIM n=20, Global Ax n=0, GAS n=9, GAS–10pt n=0, Klein Bell ADL n=0, LASIS n=0, SF-36 n=24, MAL n=5, MAL-5 n=0, MAL-28 n=1, MI n=6, NHPT n=10, OHS n=2, PDS/CBS n=0, RMA n=5, RMA-UL n=6, SA-SIP n=4, SIS n=10, UL MAS n=10)

*See Supplementary Table 2 for reasons for full text exclusions. ARAT = Action Research Arm Test, ArmA = Arm Activity Measure, AQoL = Assessment of Quality of Life, BI = Barthel Index, CMSA = Chedoke-McMaster Stroke Assessment, DAS = Disability Assessment Scale, EQ-5D = EuroQol – 5 dimension, FAT = Frenchay Arm Test, mFAT = modified Frenchay Arm Test, FIM = Functional Independence Measure, GAS = Goal Attainment Scale, GAS – 10pt = Goal Attainment Scale – 10 point, Global Ax = Global Assessment Scale, KleinBell ADL = Klein-Bell Activities of Daily Living scale, LASIS = Leeds Adult Spasticity Impact Scale, SF-36 = Medical Outcome Study 36-Item Short-Form Health Survey, MAL = Motor Activity Log, MAL-5 = Motor Activity Log - 5, MAL-28 = Motor Activity Log - 28, MI = Motricity Index, NHPT = Nine Hole Peg Test, OHS = Oxford Handicap Scale, PDS/CBS = Patient Disability Scale / Carer Burden Scale, RMA = Rivermead Motor Assessment, RMA-UL = Rivermead Motor Assessment - Upper Limb, SA-SIP = Stroke-Adapted Version of the Sickness Impact Profile, SIS = Stroke Impact Scale, UL MAS = Upper Limb Motor Assessment Scale.

**Fig 1. PRISMA flow chart.**

these are detailed in Table 2; sample sizes were commonly small (range n = 5 to n = 148,367; mean = 2335.24 (SD = 14,431.79); median = 90), with less than 100 in over half of studies (56%) and only n = 5 studies including greater than 10 000 participants. The number of studies evaluating each measurement tool varied, ranging from n = 1 study investigating the Motor Activity Log-28 (MAL-28), to n = 23 for the Medical Outcome Study 36-Item Short-Form Health Survey (SF-36). Participants with upper limb spasticity were specifically identified in n = 15 studies in total (across n = 9 of the included n = 22 measurement tools).

## Characteristics of each measurement tool

The number of studies examining each measurement tool is presented, together with findings for all participants and then for participants with upper limb spasticity. The synthesis of

**Table 2. Characteristics of included studies.**

| Studies included | Measurement tool | Summary of study participants | Psychometric property tested |
|---|---|---|---|
| Adams et al., (1997) [25] | RMA | Diagnosis = Stroke | Structural validity |
| | RMA-UL | Time since diagnosis (mo) = greater than 6 | |
| | | n = 83 | |
| | | Age (yr), mean (SD) = Grp 1: 75.39 (6.41), Grp 2: 56.54 (5.73), | |
| | | Grp 3: 56.33 (5.95) | |
| | | Sex, number male (%) = Grp 1: (51), Grp 2: (62), Grp 3: (54) | |
| | | Sample included people with spasticity = not reported | |
| Adams et al., (1997) [26] | RMA | Diagnosis = Stroke | Structural validity |
| | RMA-UL | Time since diagnosis (mo) = less than 6 | |
| | | n = 51 | |
| | | Age (yr), mean (SD) = 74.37 (9.38) | |
| | | Sex, number male (%) = 24 (47) | |
| | | Sample included people with spasticity = not reported | |
| Alderman et al., (2001) [27] | EQ-5D | Diagnosis = Traumatic Brain Injury n = 29, Stroke n = 11 | Construct validity |
| | | Time since diagnosis (mo) = greater than 6 | |
| | | n = 11 | |
| | | Age (yr), mean (range) = 39 (19–66) | |
| | | Sex, number male (%) = 42 (81) | |
| | | Sample included people with spasticity = not reported | |
| Ali et al., (2013) [28] | BI | Diagnosis = Stroke | Construct validity |
| | | Time since diagnosis (mo) = less than 6 | |
| | | n = 3787 | |
| | | Age (yr), mean (median IQR) = 71 (60–78) | |
| | | Sex, number male (%) = 2715 (55) | |
| | | Sample included people with spasticity = not reported | |
| Anderson et al., (1996) [29] | SF-36 | Diagnosis = Stroke | Internal consistency |
| | | Time since diagnosis (mo) = greater than 6 | Construct validity |
| | | n = 90 | |
| | | Age (yr), mean (SD) = 72 (12) | |
| | | Sex, number male (%) = 48 (53) | |
| | | Sample included people with spasticity = not reported | |
| Ashford et al., (2015) [30] | ArmA | Diagnosis = Mixed (Stroke n = 15, TBI n = 1) | Content validity |
| | | Time since diagnosis (mo) = greater than 6 | |
| | | n = 16 | |
| | | Age (yr), mean (SD) = 54.5 (15.7) | |
| | | Sex number male (%) = 9 (56) | |
| | | Sample included people with spasticity = yes | |
| Ashford et al., (2016) [31] | ArmA | Diagnosis = Mixed (Stroke n = 48, TBI n = 28, MS n = 6, other n = 10) | Structural validity |
| | | Time since diagnosis (mo) = not reported | |
| | | n = 92 | |
| | | Age (yr), mean (SD) = 44.5 (16.7) | |
| | | Sex number male (%) = 54 (59) | |
| | | Sample included people with spasticity = yes | |

(*Continued*)

**Table 2.** (*Continued*)

| Studies included | Measurement tool | Summary of study participants | Psychometric property tested |
|---|---|---|---|
| Ashford et al., (2014) [32] | ArmA | Diagnosis = Mixed (Stroke n = 30, MS n = 4, TBI n = 22, other n = 2) | Responsiveness |
| | | Time since diagnosis (mo) = not reported | |
| | | n = 58 | |
| | | Age (yr), mean (SD) = 47 (17.5) | |
| | | Sex number male (%) = 32 (55) | |
| | | Sample included people with spasticity = yes | |
| Ashford et al., (2013) [33] | ArmA | Diagnosis = Stroke | Content validity |
| | | Time since diagnosis (*mo*) = not given | |
| | | n = 46 (clinicians), 26 (patient, carers) | |
| | | Age (*yr*), median (range) = 48.5 (30–64) (patients) | |
| | | Sex, number male (%) = 8 (62) (patients) | |
| | | Sample included people with spasticity = yes | |
| Ashford et al., (2013) [34] | ArmA | Diagnosis = Mixed (Stroke n = 48, TBI n = 28, MS n = 6, other n = 10) | Internal consistency |
| | | Time since diagnosis (*mo*) = not reported | Reliability |
| | | n = 92 | Structural validity |
| | | Age (*yr*), mean (SD) = 44.5 (16.7) | Construct validity |
| | | Sex, number male (%) = 54 (59) | Responsiveness |
| | | Sample included people with spasticity = yes | Interpretability |
| Barer & Murphy (1993) [35] | BI (C&W) | Diagnosis = Stroke | Structural validity |
| | | Time since diagnosis (*mo*) = less than 6 | Construct validity |
| | | n = 730 | Responsiveness |
| | | Age (*yr*), mean (SD) = 73.2 (not given) | |
| | | Sex number male (%) = 336 (46) | |
| | | Sample included people with spasticity = not reported | |
| Barton et al., (2008) [36] | EQ-5D | Diagnosis = Stroke | Construct validity |
| | | Time since diagnosis (*mo*) = greater than 6 | |
| | | n = 62 | |
| | | Age ≥ 45 years | |
| | | Sex (all sample, not only Stroke), number male (%) = 865 (46.4) | |
| | | Sample included people with spasticity = not reported | |
| Barton et al., (2008) [37] | EQ-5D | Diagnosis = Stroke | Construct validity |
| | | Time since diagnosis (*mo*) = not reported | Interpretability |
| | | n = 57 | |
| | | Age (all sample, not only Stroke) (*yr*), mean (range) = 64.7 (45–99) | |
| | | Sex (all sample, not only Stroke), number male (%) = 835 (44.8) | |
| | | Sample included people with spasticity = not reported | |
| Beebe & Lang (2009) [38] | ARAT | Diagnosis = Stroke | Construct validity |
| | NHPT | Time since diagnosis (*mo*) = less than 6 | Responsiveness |
| | | n = 33 | |
| | | Age (*yr*), mean (SD) = 53.9 (10.2) | |
| | | Sex, number male (%) = 19 (58) | |
| | | Sample included people with spasticity = yes | |

(*Continued*)

**Table 2.** (Continued)

| Studies included | Measurement tool | Summary of study participants | Psychometric property tested |
|---|---|---|---|
| Benedict et al., (2011) [39] | NHPT | Diagnosis = Multiple Sclerosis | Construct validity |
| | | Time since diagnosis (mo) = not reported | |
| | | n = 211 | |
| | | Age (yr), mean (SD) = 46.2 (8.9) | |
| | | Sex, number male (%) = 32 (27) | |
| | | Sample included people with spasticity = not reported | |
| Bohannon (1999) [40] | MI | Diagnosis = Stroke | Internal consistency |
| | | Time since diagnosis (mo) = less than 6 | Construct validity |
| | | n = 10 | |
| | | Age (yr), mean (range) = 66.7 (46–81) | |
| | | Sex, number male (%) = not given | |
| | | Sample included people with spasticity = not reported | |
| Bovend'Eerdt et al., (2011) [41] | GAS | Diagnosis = Mixed (Stroke n = 27, TBI n = 1, MS n = 1) | Reliability |
| | | Time since diagnosis (mo) = less than 6 | Measurement error |
| | | n = 29 | |
| | | Age (yr), mean (SD) = 50.28 (13.88) | |
| | | Sex, number male (%) = 18 (62) | |
| | | Sample included people with spasticity = not reported | |
| Brashear et al., (2002) [42] | DAS | Diagnosis = Stroke | Reliability |
| | | Time since diagnosis (mo) = greater than 6 | Content validity |
| | | n = 10 raters | |
| | | Age (yr), mean (SD) = 59.9 (16.17) | |
| | | Sex, number male (%) = 5 (56) | |
| | | Sample included people with spasticity = yes | |
| Brock et al., (2009) [43] | GAS | Diagnosis = Stroke | Construct validity |
| | | Time since diagnosis (mo) = less than 6 | |
| | | n = 45 patients 23 carers | |
| | | Age (yr), median (range) = 66 (35–87) | |
| | | Sex, number male (%) = (56) | |
| | | Sample included people with spasticity = not reported | |
| Brown et al., (2015) [44] | FIM | Diagnosis = Stroke | Construct validity |
| | | Time since diagnosis (mo) = less than 6 | Interpretability |
| | | n = 148 367 | |
| | | Age (yr), mean (SD) = 70.6 (13.1) | |
| | | Sex, number male (%) = 71,726 (48) | |
| | | Sample included people with spasticity = not reported | |
| Burridge et al., (2009) [45] | ARAT | Diagnosis = Stroke | Construct validity |
| | | Time since diagnosis (mo) = greater than 6 | |
| | | n = 17 | |
| | | Age (yr), mean (SD) = 57 (13.4) | |
| | | Sex, number male (%) = 11 (65) | |
| | | Sample included people with spasticity = yes | |

(*Continued*)

**Table 2.** (Continued)

| Studies included | Measurement tool | Summary of study participants | Psychometric property tested |
|---|---|---|---|
| Carr et al., (1985) [46] | UL-MAS | Diagnosis = Stroke | Reliability |
| | | Time since diagnosis (mo) = less than 6 | Content validity |
| | | n = 5 | |
| | | Age (yr), mean (range) = 65 (55–78) | |
| | | Sex, number male (%) = 1 (20) | |
| | | Sample included people with spasticity = not reported | |
| Chen et al., (2012) [47] | MAL | Diagnosis = Stroke | Measurement error |
| | | Time since diagnosis (mo) = 3–9 | Interpretability |
| | | n = 116 | |
| | | Age (yr), range = Intervention grp 60.98 (13.47) | |
| | | Control grp 63.26 (12.56) | |
| | | Sex, number male (%) = Intervention grp 69 (65) | |
| | | Control grp 73 (63) | |
| | | Sample included people with spasticity = not reported | |
| Collin & Wade (1990) [48] | MI | Diagnosis = Stroke | Reliability |
| | RMA–UL | Time since diagnosis (mo) = less than 6 | Construct validity |
| | | n = 20 (reliability), n = 14 (concurrent validity) | |
| | | Age (yr) mean (range) = 56.1 (15–77) | |
| | | Sex number male (%) = 24 (67) | |
| | | Sample included people with spasticity = not reported | |
| Collin et al., (1988) [49] | BI (C&W) | Diagnosis = Mixed (Stroke n = 13, Traumatic Brain Injury n = 11, other n = 1) | Reliability |
| | | Time since diagnosis (mo) = less than 6 | Content validity |
| | | n = 25 | |
| | | Age (yr), range = 12–66 | |
| | | Sex number male (%) = 124 (52) | |
| | | Sample included people with spasticity = not reported | |
| Corrigan et al., (1997) [50] | FIM | Diagnosis = Traumatic Brain Injury | Construct validity |
| | | Time since diagnosis (mo) = greater than 6 | |
| | | n = 95 | |
| | | Age (yr), mean (SD) = 35.2 (not given) | |
| | | Sex, number male (%) = 67 (70) | |
| | | Sample included people with spasticity = not reported | |
| Costelloe et al., (2008) [51] | NHPT | Diagnosis = Multiple Sclerosis | Construct validity |
| | | Time since diagnosis (mo) = not reported | Interpretability |
| | | n = 150 | |
| | | Age (yr), mean (SD) = not given | |
| | | Sex, number male (%) = not given | |
| | | Sample included people with spasticity = not reported | |
| Cullen et al., (2014) [52] | FIM | Diagnosis = Traumatic Brain Injury | Construct validity |
| | | Time since diagnosis (mo) = greater than 6 | |
| | | n = 59 | |
| | | Age (yr), mean (SD) = drivers 49.77 (15.25) | |
| | | non-driver 51.42 (15.73) | |
| | | Sex, number male (%) = driver 28 (80) non-driver 19 (79) | |
| | | Sample included people with spasticity = not reported | |

(*Continued*)

**Table 2.** (Continued)

| Studies included | Measurement tool | Summary of study participants | Psychometric property tested |
|---|---|---|---|
| Cuthbert et al., (2015) [53] | FIM | Diagnosis = Traumatic Brain Injury | Construct validity |
| | | Time since diagnosis (mo) = greater than 6 | |
| | | n = 64081 | |
| | | Age (yr), mean = 76% less than 80 | |
| | | Sex, number male (%) = 41204 (64.3) | |
| | | Sample included people with spasticity = not reported | |
| Dang et al., (2011) [54] | CMSA | Diagnosis = Stroke | Construct validity |
| | | Time since diagnosis (mo) = less than 6 | |
| | | n = 74 | |
| | | Age (yr), mean (SD) = 65.3 (12.4) | |
| | | Sex, number male (%) = 48 (65) | |
| | | Sample included people with spasticity = not reported | |
| Demeurisse et al., (1980) [55] | MI | Diagnosis = Stroke | Content validity |
| | | Time since diagnosis (mo) = less than 6 | |
| | | n = 100 | |
| | | Age (yr), mean (SD) = 69 (not reported) | |
| | | Sex, number male (%) = 59 (59) | |
| | | Sample included people with spasticity = not reported | |
| Dennis et al., (2000) [56] | BI (C&W) | Diagnosis = Stroke | Construct validity |
| | | Time since diagnosis (mo) = greater than 6 | |
| | | n = 417 | |
| | | Age (yr), mean (SD) = 64.6 (not given) | |
| | | Sex number male (%) = not reported | |
| | | Sample included people with spasticity = not reported | |
| De Weerdt et al., (1985) [57] | ARAT | Diagnosis = Stroke | Construct validity |
| | | Time since diagnosis (mo) = less than 6 | Responsiveness |
| | | n = 53 | |
| | | Age (yr), mean (SD) = 68.6 (9.3) | |
| | | Sex, number male (%) = 25 (47) | |
| | | Sample included people with spasticity = not reported | |
| Doan et al., (2012) [58] | DAS | Diagnosis = Stroke | Construct validity |
| | EQ-5D | Time since diagnosis (mo) = greater than 6 | |
| | SA-SIP30 | n = 279 | |
| | | Age (yr), mean (range) = 58.2 (21–88) | |
| | | Sex, number male (%) = 150 (54) | |
| | | Sample included people with spasticity = yes | |
| Doig et al., (2010) [59] | GAS | Diagnosis = Traumatic Brain Injury | Construct validity |
| | | Time since diagnosis (mo) = greater than 6 | Responsiveness |
| | | n = 14 | |
| | | Age (yr), range = 18–57 | |
| | | Sex, number male (%) = 12 (86) | |
| | | Sample included people with spasticity = not reported | |

(*Continued*)

**Table 2.** (Continued)

| Studies included | Measurement tool | Summary of study participants | Psychometric property tested |
|---|---|---|---|
| Dorman et al., (1999) [60] | SF-36 | Diagnosis = Stroke | Construct validity |
| | EQ-5D | Time since diagnosis (mo) = greater than 6 | Interpretability |
| | | n = 531 | |
| | | Age (yr), mean (SD) = not reported | |
| | | Sex, number male (%) = not reported | |
| | | Sample included people with spasticity = not reported | |
| Dorman et al., (1998) [61] | SF-36 | Diagnosis = Stroke | Internal consistency |
| | EQ-5D | Time since diagnosis (mo) = greater than 6 | Reliability |
| | | n = 209 | |
| | | Age (yr), mean = 70 | |
| | | Sex, number male (%) = 147 (54) | |
| | | Sample included people with spasticity = not reported | |
| Dorman et al., (1997) [62] | EQ-5D | Diagnosis = Stroke | Construct validity |
| | | Time since diagnosis (mo) = not reported | |
| | | n = 152 | |
| | | Age % of sample by group <50 = 5%, 50–70 = 46%, >70 = 49%. | |
| | | Sex, number male (%) = not reported | |
| | | Sample included people with spasticity = not reported | |
| Dromerick et al., (2006) [63] | ARAT | Diagnosis = Stroke | Construct validity |
| | MAL | Time since diagnosis (mo) = less than 6 | Interpretability |
| | | n = 39 | |
| | | Age (yr), mean (SD) = 64.54 (14.13) | |
| | | Sex, number male (%) = 17 (44) | |
| | | Sample included people with spasticity = not reported | |
| Duncan et al., (2003) [64] | SIS | Diagnosis = Stroke | Content validity |
| | | Time since diagnosis (mo) = less than 6 | Structural validity |
| | | n = 696 | |
| | | Age (yr), mean (SD) = 68.6 (12.5) | |
| | | Sex, number male (%) = 386 (55) | |
| | | Sample included people with spasticity = not reported | |
| Duncan et al., (2002) [65] | SIS | Diagnosis = Stroke | Reliability |
| | | Time since diagnosis (mo) = less than 6 | Construct validity |
| | | n = 287 | |
| | | Age (yr), mean (SD) = 72.6 (10), 59.8 (15.5) | |
| | | Sex, number male (%) = 135 (47), 78 27.2 | |
| | | Sample included people with spasticity = not reported | |
| Duncan et al., (2005) [66] | SIS | Diagnosis = Stroke | Internal consistency |
| | | Time since diagnosis (mo) = less than 6 | Reliability |
| | | n = 26 | |
| | | Age (yr), mean (SD) = mail sample 68.48 (11.4) | |
| | | telephone sample 68.84 (12.2) | |
| | | Sex, number male (%) = mail sample 219 (97.8) | |
| | | telephone sample 230 (98.3) | |
| | | Sample included people with spasticity = not reported | |

(*Continued*)

**Table 2.** (Continued)

| Studies included | Measurement tool | Summary of study participants | Psychometric property tested |
|---|---|---|---|
| Duncan et al., (1997) [67] | SF-36 | Diagnosis = Stroke | Construct validity |
| | | Time since diagnosis *(mo)* = greater than 6 | |
| | | n = 200 | |
| | | Age *(yr)*, mean (SD) = 63 (13) | |
| | | Sex, number male (%) = 164 (54) | |
| | | Sample included people with spasticity = not reported | |
| Duncan et al., (1999) [68] | SIS | Diagnosis = Stroke | Content validity |
| | | Time since diagnosis *(mo)* = less than 6 | |
| | | n = 91 | |
| | | Age *(yr)*, mean (SD) = minor stroke 69.2 (10.1) | |
| | | moderate stroke 71.9 (11.7) | |
| | | Sex, number male (%) = 42 (46) | |
| | | Sample included people with spasticity = not reported | |
| Edwards et al., (2006) [69] | SA-SIP30 | Diagnosis = Stroke | Construct validity |
| | | Time since diagnosis *(mo)* = greater than 6 | |
| | | n = 219 | |
| | | Age *(yr)*, mean (SD) = 64.74 (15.87) | |
| | | Sex, number male (%) = 94 (43) | |
| | | Sample included people with spasticity = not reported | |
| Egan et al., (2014) [70] | FIM | Diagnosis = Stroke | Construct validity |
| | | Time since diagnosis *(mo)* = greater than 6 | |
| | | n = 55 | |
| | | Age *(yr)*, mean (SD) = 64.8 (13.3) | |
| | | Sex, number male (%) = 39 (58) | |
| | | Sample included people with spasticity = not reported | |
| Eriksson et al., (2013) Eriksson, Baum [71] | SIS | Diagnosis = Stroke | Construct validity |
| | | Time since diagnosis *(mo)* = greater than 6 | Interpretability |
| | | n = 116 | |
| | | Age *(yr)*, mean (SD) = 62.4 (12.7) | |
| | | Sex number male (%) = 56 (48) | |
| | | Sample included people with spasticity = not reported | |
| Filiatrault et al., (1991) [72] | BI | Diagnosis = Stroke | Construct validity |
| | | Time since diagnosis *(mo)* = less than 6 | Responsiveness |
| | | n = 18 | |
| | | Age *(yr)*, mean (SD) = 52.2 (13.5) | |
| | | Sex number male (%) = 12 (67) | |
| | | Sample included people with spasticity = not reported | |
| Fisk et al., (2005) [73] | EQ-5D | Diagnosis = Multiple Sclerosis | Construct validity |
| | | Time since diagnosis = not given | |
| | | n = 187 | |
| | | Age *(yr)*, mean (SD) = 51 (10) | |
| | | Sex, number male (%) = 47 (25) | |
| | | Sample included people with spasticity = not reported | |

(*Continued*)

**Table 2.** (*Continued*)

| Studies included | Measurement tool | Summary of study participants | Psychometric property tested |
|---|---|---|---|
| Findler et al., (2001) [74] | SF-36 | Diagnosis = Traumatic Brain Injury | Construct validity |
| | | Time since diagnosis (*mo*) = greater than 6 | |
| | | n = 326 | |
| | | Age (yr), mean (SD) = 41.7 (10.8) mild, 35.7 (9.8) moderate-severe | |
| | | Sex, number male (%) = 130 (88) | |
| | | Sample included people with spasticity = not reported | |
| Fleming et al., (2014) [75] | ARAT | Diagnosis = Stroke | Construct validity |
| | | Time since diagnosis (*mo*) = greater than 6 | Interpretability |
| | | n = 33 | |
| | | Age (*yr*), mean (SD) = 61.5 (14.2) | |
| | | Sex, number male (%) = 20 (61) | |
| | | Sample included people with spasticity = yes | |
| Freeman et al., (2000) [76] | SF-36 | Diagnosis = Multiple Sclerosis | Internal consistency |
| | | Time since diagnosis (*mo*) = greater than 6 | Construct validity |
| | | n = 149 | Responsiveness |
| | | Age (yr), mean (SD) = 44.6 (10.8) | Interpretability |
| | | Sex, number male (%) = (32) | |
| | | Sample included people with spasticity = not reported | |
| Freeman et al., (1996) [77] | SF-36 | Diagnosis = Multiple Sclerosis | Construct validity |
| | | Time since diagnosis (*mo*) = greater than 6 | Interpretability |
| | | n = 50 | |
| | | Age (*yr*), mean (SD) = 44.8 (9.8) | |
| | | Sex, number male (%) = 21 (42) | |
| | | Sample included people with spasticity = not reported | |
| Gillard et al., (2015) [78] | EQ-5D | Diagnosis = Stroke | Construct validity |
| | | Time points since diagnosis (*mo*) = greater than 6 | |
| | | n = 460 | |
| | | Age (*yr*), mean (SD) = 67 (14) | |
| | | Sex, number male (%) = 241 (52) | |
| | | Sample included people with spasticity = yes | |
| Goodkin et al., (1988) [79] | NHPT | Diagnosis = Multiple Sclerosis | Construct validity |
| | | Time since diagnosis (*mo*) = greater than 6 | Interpretability |
| | | n = Exp 68, Control 21 | |
| | | Age (*yr*), mean (SD) = Exp 47.16 (11.3) Control 45.24 (16.50) | |
| | | Sex number male (%) = Exp 25 (37) Control 7 (33) | |
| | | Sample included people with spasticity = not reported | |
| Gowland 1990 [80] | CMSA | Diagnosis = Stroke | Content validity |
| | | Time since diagnosis (*mo*) = not reported | |
| | | n = not reported | |
| | | Age (*yr*), mean (range) = not reported | |
| | | Sex, number male (%) = not reported | |
| | | Sample included people with spasticity = not reported | |

(*Continued*)

**Table 2.** (Continued)

| Studies included | Measurement tool | Summary of study participants | Psychometric property tested |
|---|---|---|---|
| Gowland et al., (1993) [81] | CMSA | Diagnosis = Stroke | Reliability |
| | | Time since diagnosis *(mo)* = less than 6 | Construct validity |
| | | n = 32 | Responsiveness |
| | | Age *(yr)*, mean (range) = 64, (18–86) | |
| | | Sex, number male (%) = 14 (44) | |
| | | Sample included people with spasticity = not reported | |
| Grant et al., (2014) [82] | FIM | Diagnosis = Stroke | Construct validity |
| | | Time since diagnosis *(mo)* = less than 6 | |
| | | n = 11983 | |
| | | Age *(yr)*, median (25th, 75th percentile) = 72 (61, 81) | |
| | | Sex, number male (%) = 6581 (55) | |
| | | Sample included people with spasticity = not reported | |
| Green et al., (2001) [83] | BI (C&W) | Diagnosis = Stroke | Reliability |
| | | Time since diagnosis *(mo)* = greater than 6 | Measurement error |
| | | n = 22 | |
| | | Age *(yr)*, mean (SD) = 71.6 (6.8) | |
| | | Sex number male (%) = 16 (73) | |
| | | Sample included people with spasticity = not reported | |
| Guilfoyle et al., (2010) [84] | SF-36 | Diagnosis = Traumatic Brain Injury | Internal consistency |
| | | Time since diagnosis *(mo)* = mixed, mean less than 6 | Structural validity |
| | | n = 453 | Construct validity |
| | | Age *(yr)*, mean (SD) = 36.6 (16.1) | Interpretability |
| | | Sex, number male (%) = 392 (76.3) | |
| | | Sample included people with spasticity = not reported | |
| Hagen et al., (2003) [85] | SF-36 | Diagnosis = Stroke | Internal consistency |
| | | Time since diagnosis *(mo)* = less than 6 | Construct validity |
| | | n = 136 | Responsiveness |
| | | Age *(yr)*, mean (SD) = 70 (11) | Interpretability |
| | | Sex, number male (%) = 69 (51) | |
| | | Sample included people with spasticity = not reported | |
| Hall et al., (1993) [86] | FIM | Diagnosis = Traumatic Brain Injury | Structural validity |
| | | Time since diagnosis *(mo)* = less than 6 | Construct validity |
| | | n = 332 | Interpretability |
| | | Age *(yr)*, mean (SD) = 34.5 (16) | |
| | | Sex, number male (%) = 259 (78) | |
| | | Sample included people with spasticity = not reported | |
| Hamilton & Granger (1994) [87] | FIM | Diagnosis = Stroke | Reliability |
| | | Time since diagnosis *(mo)* = less than 6 | |
| | | n = 1018 | |
| | | Age *(yr)*, mean (SD) = 71 (12) | |
| | | Sex, number male (%) = 478 (47) | |
| | | Sample included people with spasticity = not reported | |

(*Continued*)

**Table 2.** (Continued)

| Studies included | Measurement tool | Summary of study participants | Psychometric property tested |
|---|---|---|---|
| Harris & Eng (2007) [88] | MAL | Diagnosis = Stroke | Construct validity |
| | | Time since diagnosis *(mo)* = greater than 6 | |
| | | n = 93 | |
| | | Age *(yr)*, mean (SD) = 68.7 (9.4) | |
| | | Sex, number male (%) = 61 (65) | |
| | | Sample included people with spasticity = yes | |
| Hawthorne et al., (2009) [89] | AQoL | Diagnosis = Traumatic Brain Injury | Construct validity |
| | | Time since diagnosis *(mo)* = greater than 6 | |
| | | n = 56 | |
| | | Age *(yr)*, mean (SD) = 39 (15) | |
| | | Sex, number male (%) = 40 (71) | |
| | | Sample included people with spasticity = not reported | |
| Hawthorne et al., (1999) [90] | AQoL | Diagnosis = Mixed (medical and musculoskeletal diagnoses, healthy samples) | Content validity |
| | | Time since diagnosis *(mo)* = less than 6 | |
| | | n = 255 | |
| | | Age *(yr)*, range = ≤29–70+ | |
| | | Sex, number male (%) = 121 (47) | |
| | | Sample included people with spasticity = not reported | |
| Heinemann et al., (1997) [91] | FIM | Diagnosis = Traumatic Brain Injury | Construct validity |
| | | Time since diagnosis *(mo)* = less than 6 | |
| | | n = 129 | |
| | | Age *(yr)*, mean (SD) = 37.4 (19.5) | |
| | | Sex, number male (%) = (71) | |
| | | Sample included people with spasticity = not reported | |
| Heinemann et al., (1993) [92] | FIM | Diagnosis = Mixed (Stroke n = 10092) | Structural validity |
| | | Time since diagnosis *(mo)* = less than 6 | |
| | | n = 10092 | |
| | | Age *(yr)*, mean (SD) = 62.1 (not given) whole sample | |
| | | Sex, number male (%) = 5349 (53) whole sample | |
| | | Sample included people with spasticity = not reported | |
| Heinemann et al., (1994) [93] | FIM | Diagnosis = Mixed (Stroke n = 9961) | Structural validity |
| | | Time since diagnosis *(mo)* = less than 6 | |
| | | n = 9961 | |
| | | Age *(yr)*, mean (SD) = 70.4 (not reported) | |
| | | Sex, number male (%) = 4781 (48) | |
| | | Sample included people with spasticity = not reported | |
| Heller et al., (1987) [94] | mFAT | Diagnosis = Stroke | Reliability |
| | NHPT | Time since diagnosis *(mo)* = greater than 6 | |
| | | n = 10 | |
| | | Age (yr) = not provided | |
| | | Sex, number male (%) = not reported | |
| | | Sample included people with spasticity = not reported | |

(*Continued*)

**Table 2.** (Continued)

| Studies included | Measurement tool | Summary of study participants | Psychometric property tested |
|---|---|---|---|
| Heller et al., (1987) [94] | mFAT | Diagnosis = Stroke | Construct validity |
| | NHPT | Time since diagnosis *(mo)* = less than 6 | Interpretability |
| | | n = 56 | |
| | | Age (yr) = 68.1 (11.4) | |
| | | Sex, number male (%) = 24 (43) | |
| | | Sample included people with spasticity = not reported | |
| Hermann et al., (1996) [95] | SF-36 | Diagnosis = Multiple Sclerosis | Construct validity |
| | | Time since diagnosis *(mo)* = greater than 6 | |
| | | n = 85 | |
| | | Age *(yr)*, mean (SD) = 44.6 () | |
| | | Sex, number male (%) = 20 (23) | |
| | | Sample included people with spasticity = not reported | |
| Hobart et al., (2002) [96] | SF-36 | Diagnosis = Stroke | Internal consistency |
| | | Time since diagnosis *(mo)* = less than 6 | Structural validity |
| | | n = 177 | Interpretability |
| | | Age *(yr)*, mean (SD) = 62 (13) | |
| | | Sex, number male (%) = 126 (71) | |
| | | Sample included people with spasticity = not reported | |
| Houlden et al., (2006) [97] | FIM | Diagnosis = Mixed (Stroke n = 261, Traumatic Brain Injury n = 107) | Responsiveness |
| | BI (C&W) | Time since diagnosis *(mo)* = less than 6 | Interpretability |
| | | n = 368 | |
| | | Age *(yr)*, mean (SD) = whole sample not reported | |
| | | Sex number male (%) = 259 (63) | |
| | | Sample included people with spasticity = not reported | |
| Jacob-Lloyd et al., (2005) [98] | MI | Diagnosis = Stroke | Construct validity |
| | NHPT | Time since diagnosis *(mo)* = less than 6 | Responsiveness |
| | | n = 58 | Interpretability |
| | | Age *(yr)* number (%) = 47 (85) older than 60 | |
| | | Sex, number male (%) = 31 (53) | |
| | | Sample included people with spasticity = not reported | |
| Jenkinson et al., (2013) [99] | SIS | Diagnosis = Stroke | Internal consistency |
| | | Time since diagnosis *(mo)* = greater than 6 | Structural validity |
| | | n = 73 | |
| | | Age *(yr)* range = 18 - >75 | |
| | | Sex, number male (%) = 88 (58) | |
| | | Sample included people with spasticity = not reported | |
| Johnson & Selfe (2004) [100] | UL-MAS | Diagnosis = Stroke | Internal consistency |
| | | Time since diagnosis *(mo)* = less than 6 | |
| | | n = 26 | |
| | | Age *(yr)* mean (SD) = 77 (9) | |
| | | Sex, number male (%) = 13 (50) | |
| | | Sample included people with spasticity = not reported | |

*(Continued)*

**Table 2.** (*Continued*)

| Studies included | Measurement tool | Summary of study participants | Psychometric property tested |
|---|---|---|---|
| Jones (1998) [101] | RMA | Diagnosis = Stroke | Construct validity |
| | | Time since diagnosis *(mo)* = less than 6 | |
| | | n = 29 | |
| | | Age *(yr)* mean (SD) = 66 (9.4) | |
| | | Sex, number male (%) = 13 (50) | |
| | | Sample included people with spasticity = not reported | |
| Joyce et al., (1994) [102] | GAS | Diagnosis = Traumatic Brain Injury | Reliability |
| | | Time since diagnosis *(mo)* = less than 6 | Content validity |
| | | n = 16 | Construct validity |
| | | Age *(yr)* mean (range) = 27 (17–49) | |
| | | Sex, number male (%) = 9 (56) | |
| | | Sample included people with spasticity = not reported | |
| Khan et al., (2013) [103] | UL-MAS | Diagnosis = Stroke | Structural validity |
| | | Time since diagnosis *(mo)* = less than 6 | Construct validity |
| | | n = 481 | |
| | | Age *(yr)* range = 18–101 | |
| | | Sex, number male (%) = 255 (53) | |
| | | Sample included people with spasticity = not reported | |
| Khan et al., (2008) [104] | GAS | Diagnosis = Multiple Sclerosis | Construct validity |
| | | Time since diagnosis *(mo)* = greater than 6 | Responsiveness |
| | | n = 24 (203 goals) | |
| | | Age *(yr)* mean (SD) = 52 (8.3) | |
| | | Sex, number male (%) = 10 (42) | |
| | | Sample included people with spasticity = not reported | |
| Keith et al., (1987) [105] | FIM | Diagnosis = not reported | Content validity |
| | | Time since diagnosis *(mo)* = not reported | |
| | | n = not reported | |
| | | Age *(yr)*, mean (SD) = not reported | |
| | | Sex, number male (%) = not reported | |
| | | Sample included people with spasticity = not reported | |
| Kohn et al., (2014) [106] | EQ-5D | Diagnosis = Multiple Sclerosis | Construct validity |
| | | Time since diagnosis *(mo)* = greater than 6 | Responsiveness |
| | | n = 3044 | |
| | | Age *(yr)*, mean (SD) = 56.8 (9.9) | |
| | | Sex, number male (%) = 600 (20) | |
| | | Sample included people with spasticity = not reported | |
| Kuspinar et al (2014) [107] | EQ-5D | Diagnosis = MS | Construct validity |
| | | Time since diagnosis *(mo)* = greater than 6 | |
| | | n = 189 | |
| | | Age *(yr)*, mean (SD) = 43 (10) | |
| | | Sex, number male (%) = 49 (26) | |
| | | Sample included people with spasticity = not reported | |

(*Continued*)

**Table 2.** (*Continued*)

| Studies included | Measurement tool | Summary of study participants | Psychometric property tested |
|---|---|---|---|
| Kuspinar & Mayo (2013) [108] | EQ-5D | Diagnosis = Multiple Sclerosis | Content validity |
| | | Time since diagnosis (*mo*) = greater than 6 | Construct validity |
| | | n = 185 | |
| | | Age (*yr*), mean (SD) = 42.8 (10) | |
| | | Sex, number male (%) = 48 (26) | |
| | | Sample included people with spasticity = not reported | |
| Kuys et al., (2009) [109] | FIM | Diagnosis = Stroke | Construct validity |
| | UL-MAS | Time since diagnosis (*mo*) = less than 6 | |
| | | n = 105 | |
| | | Age (*yr*) median = 70 (13) | |
| | | Sex, number male (%) = 64 (53) | |
| | | Sample included people with spasticity = not reported | |
| Kwon et al., (2006) [110] | SIS | Diagnosis = Stroke | Construct validity |
| | | Time since diagnosis (*mo*) = less than 6 | Interpretability |
| | | n = 95 | |
| | | Age (*yr*) median = 70 (13) | |
| | | Sex, number male (%) = 64 (53) | |
| | | Sample included people with spasticity = not reported | |
| Kwon et al., (2004) [111] | BI | Diagnosis = Stroke | Construct validity |
| | | Time since diagnosis (*mo*) = less than 6 | |
| | | n = 1680 | |
| | | Age (*yr*), mean (SD) = 70 (11.4) | |
| | | Sex number male (%) = 790 (47) | |
| | | Sample included people with spasticity = not reported | |
| Lai et al., (2002) [112] | SIS | Diagnosis = Stroke | Construct validity |
| | | Time since diagnosis (*mo*) = less than 6 | Interpretability |
| | | n = 81 | |
| | | Age (*yr*), mean (SD) = 76 (6.56) | |
| | | Sex number male (%) = 48 (59) | |
| | | Sample included people with spasticity = not reported | |
| Lang et al., (2008) [113] | ARAT | Diagnosis = Stroke | Interpretability |
| | | Time since diagnosis (*mo*) = less than 6 | |
| | | n = 12 | |
| | | Age (*yr*), mean (SD) = 64 (14) | |
| | | Sex, number male (%) = 21 (40) | |
| | | Sample included people with spasticity = not reported | |
| Lang et al., (2006) [114] | ARAT | Diagnosis = Stroke | Construct validity |
| | | Time since diagnosis (*mo*) = less than 6 | Responsiveness |
| | | n = 50 | |
| | | Age (*yr*), mean (SD) = 63.7 (13.6) | |
| | | Sex, number male (%) = 21 (42) | |
| | | Sample included people with spasticity = yes | |

(*Continued*)

**Table 2.** (Continued)

| Studies included | Measurement tool | Summary of study participants | Psychometric property tested |
|---|---|---|---|
| Lannin (2003) [115] | GAS | Diagnosis = mixed (Stroke, Traumatic Brain Injury) | Responsiveness |
| | | Time since diagnosis (*mo*) = greater than 6 | |
| | | n = 12 | |
| | | Age (*yr*), mean (range) = 56.5 (26–79) | |
| | | Sex, number male (%) = not reported | |
| | | Sample included people with spasticity = not reported | |
| Lannin (2004) [116] | UL-MAS | Diagnosis = Stroke | Internal consistency |
| | | Time since diagnosis (*mo*) = less than 6 | Structural validity |
| | | n = 27 | |
| | | Age (*yr*), mean (SD) = 67 (10.1) | |
| | | Sex, number male (%) = 15 (50) | |
| | | Sample included people with spasticity = not reported | |
| Lincoln & Leadbitter (1979) [117] | RMA | Diagnosis = Stroke | Content validity |
| | | Time since diagnosis (*mo*) = not reported | |
| | | n = 51 | |
| | | Age (*yr*), range = 17–65 | |
| | | Sex, number male (%) = not reported | |
| | | Sample included people with spasticity = not reported | |
| Loewen & Anderson (1988) [118] | UL-MAS | Diagnosis = Stroke | Reliability |
| | | Time since diagnosis (*mo*) = less than 6 | |
| | | n = 7 | |
| | | Age (*yr*), mean (SD) = 73.6 (8.3) | |
| | | Sex, number male (%) = 2 (29) | |
| | | Sample included people with spasticity = not reported | |
| Loewen & Anderson (1990) [119] | UL-MAS | Diagnosis = Stroke | Construct validity |
| | | Time since diagnosis (*mo*) = less than 6 | |
| | | n = 50 | |
| | | Age (*yr*), mean (SD) = 68 (10) | |
| | | Sex, number male (%) = 28 (56) | |
| | | Sample included people with spasticity = not reported | |
| Lyle (1981) [120] | ARAT | Diagnosis = Mixed (Stroke n = unknown, Traumatic Brain Injury n = unknown) | Content validity |
| | | Time since diagnosis (*mo*) = Greater than 6) | Structural validity |
| | | n = 20 | |
| | | Age (*yr*), mean (range) = 53.2 (26–72) | |
| | | Sex, number male (%) = 13 (65) | |
| | | Sample included people with spasticity = not reported | |
| Mackenzie et al., (2002) [121] | SF-36 | Diagnosis = Traumatic Brain Injury | Structural validity |
| | | Time since diagnosis (*mo*) = greater than 6 | Construct validity |
| | | n = 1197 | |
| | | Age (*yr*), range = 18–54 | |
| | | Sex, number male (%) = 790 (66) | |
| | | Sample included people with spasticity = not reported | |

(*Continued*)

**Table 2.** (*Continued*)

| Studies included | Measurement tool | Summary of study participants | Psychometric property tested |
|---|---|---|---|
| Madden et al., (2006) [122] | SF-36 | Diagnosis = Stroke | Construct validity |
| | | Time since diagnosis *(mo)* = less than 6 | Responsiveness |
| | | n = 116 | Interpretability |
| | | Age *(yr)*, mean (range) = 70 (10) | |
| | | Sex, number male (%) = 57 (49) | |
| | | Sample included people with spasticity = not reported | |
| Mahoney & Barthel (1965) [123] | BI | Diagnosis = not given | Content validity |
| | | Time since diagnosis *(mo)* = not given | |
| | | n = not given | |
| | | Age *(yr)*, mean (range) = not given | |
| | | Sex, number male (%) = not given | |
| | | Sample included people with spasticity = not reported | |
| Malec (1999) [124] | GAS | Diagnosis = Mixed (Traumatic Brain Injury n = 66, Stroke n = 15, other n = 7) | Construct validity |
| | | Time since diagnosis *(mo)* = greater than 6 (61%) | |
| | | n = 88 | |
| | | Age *(yr)*, mean (range) = 33.8 (18–69) | |
| | | Sex number male (%) = 64 (72.7) | |
| | | Sample included people with spasticity = not reported | |
| Malec et al., (1991) [125] | GAS | Diagnosis = Traumatic Brain Injury | Construct validity |
| | | Time since diagnosis *(mo)* = greater than 6 | |
| | | n = 14 | |
| | | Age *(yr)*, mean (SD) = 34.3 (12.2) | |
| | | Sex, number male (%) = not reported | |
| | | Sample included people with spasticity = not reported | |
| Miller et al., (2010) [126] | UL-MAS | Diagnosis = Stroke | Internal consistency |
| | | Time since diagnosis *(mo)* = less than 6 | Structural validity |
| | | n = 80 | Construct validity |
| | | Age *(yr)*, mean (SD) = 67.4 (15.6) | Interpretability |
| | | Sex, number male (%) = 46 (58) | |
| | | Sample included people with spasticity = not reported | |
| Moore et al., (2004) [127] | SF-36 | Diagnosis = Multiple Sclerosis | Construct validity |
| | EQ-5D | Time since diagnosis *(mo)* = greater than 6 | |
| | | n = 114 | |
| | | Age *(yr)*, mean (SD) = 45 (11) | |
| | | Sex, number male (%) = 18 (45) | |
| | | Sample included people with spasticity = not reported | |
| Moreland et al., (1993) [128] | CMSA | Diagnosis = Stroke | Content validity |
| | | Time since diagnosis *(mo)* = not reported | |
| | | n = not reported | |
| | | Age *(yr)*, median (range) = not reported | |
| | | Sex, number male (%) = not reported | |
| | | Sample included people with spasticity = not reported | |

(*Continued*)

**Table 2.** (Continued)

| Studies included | Measurement tool | Summary of study participants | Psychometric property tested |
|---|---|---|---|
| Morris et al., (2013) [129] | ARAT | Diagnosis = Stroke | Construct validity |
| | NHPT | Time since diagnosis (*mo*) = greater than 6 | Interpretability |
| | RMA–UL | n = 85 | |
| | | Age (*yr*), median (range) = 69 (36–88) | |
| | | Sex, number male (%) = 49 (58) | |
| | | Sample included people with spasticity = not reported | |
| Murrell et al., (1999) [130] | SF-36 | Diagnosis = Multiple Sclerosis | Reliability |
| | | Time since diagnosis (*mo*) = greater than 6 | |
| | | n = 22 | |
| | | Age (*yr*), mean (SD) = 52.4 (9.9) | |
| | | Sex, number male (%) = 9 (40) | |
| | | Sample included people with spasticity = not reported | |
| Nicholl et al., (2001) [131] | EQ-5D | Diagnosis = Multiple Sclerosis | Construct validity |
| | | Time points since diagnosis (*mo*) = greater than 6 | Interpretability |
| | | n = 88 | |
| | | Age (*yr*), mean (SD) = 48.97 (8.9) | |
| | | Sex, number male (%) = 24 (25) | |
| | | Sample included people with spasticity = not reported | |
| Oczkowski et al., (1993) [132] | FIM | Diagnosis = Stroke | Construct validity |
| | | Time since diagnosis (*mo*) = less than 6 | |
| | | n = 113 | |
| | | Age (*yr*), mean = 65.7 (female) 65.8 (male) | |
| | | Sex, number male (%) = 59 (52.2) | |
| | | Sample included people with spasticity = not reported | |
| O'Mahony et al., (1998) [133] | SF-36 | Diagnosis = Stroke | Interpretability |
| | | Time since diagnosis (*mo*) = not reported | |
| | | n = 104 | |
| | | Age (*yr*), mean (range) = > 45 | |
| | | Sex, number male (%) = not reported | |
| | | Sample included people with spasticity = not reported | |
| Ouellette et al., (2015) [134] | FIM | Diagnosis = Stroke | Construct validity |
| | | Time since diagnosis (*mo*) = less than 6 | |
| | | n = 407 | |
| | | Age (*yr*), mean (SD) = 68.2 (13.9) | |
| | | Sex, number male (%) = not given | |
| | | Sample included people with spasticity = not reported | |
| Peters et al., (2014) [135] | EQ-5D | Diagnosis = Stroke | Responsiveness |
| | | Time since diagnosis (*mo*) = not reported | |
| | | n = 102 | |
| | | Age (*yr*) = 78% > 55 | |
| | | Sex, number male (%) = 53 (53) | |
| | | Sample included people with spasticity = not reported | |

(*Continued*)

**Table 2.** (Continued)

| Studies included | Measurement tool | Summary of study participants | Psychometric property tested |
|---|---|---|---|
| Pickard et al., (2005) [136] | EQ-5D | Diagnosis = Stroke | Responsiveness |
| | | Time points since diagnosis *(mo)* = less than 6 | Interpretability |
| | | n = 96 | |
| | | Age *(yr)*, mean (SD) = 67 (15) | |
| | | Sex, number male (%) = 51 (52) | |
| | | Sample included people with spasticity = not reported | |
| Pickering et al., (2010) [137] | UL-MAS | Diagnosis = Stroke | Structural validity |
| | | Time since diagnosis *(mo)* = less than 6 | Interpretability |
| | | n = 25 | |
| | | Age *(yr)*, mean (SD) = 69.96 (11.97) | |
| | | Sex, number male (%) = 14 (56) | |
| | | Sample included people with spasticity = not reported | |
| Pittock et al., (2004) [138] | SF-36 | Diagnosis = Multiple Sclerosis | Construct validity |
| | | Time since diagnosis *(mo)* = greater than 6 | |
| | | n = 185 | |
| | | Age *(yr)*, mean (SD) = not given | |
| | | Sex, number male (%) = 56 (30) | |
| | | Sample included people with spasticity = not reported | |
| Poole et al., (2010) [139] | NHPT | Diagnosis = Multiple Sclerosis | Construct validity |
| | | Time since diagnosis *(mo)* = greater than 6 | |
| | | n = 56 | |
| | | Age *(yr)*, mean (SD) = 46.8 (10.48) | |
| | | Sex, number male (%) = 11 (20) | |
| | | Sample included people with spasticity = not reported | |
| Rabadi & Rabadi (2006) [140] | ARAT | Diagnosis = Stroke | Construct validity |
| | | Time since diagnosis *(mo)* = less than 6 | Responsiveness |
| | | n = 104 | |
| | | Age *(yr)*, mean (SD) = 72.0 (13) | |
| | | Sex, number male (%) = 43 (41) | |
| | | Sample included people with spasticity = not reported | |
| Rabadi & Vincent (2013) [141] | FIM | Diagnosis = Multiple Sclerosis | Construct validity |
| | | Time since diagnosis *(mo)* = greater than 6 | Responsiveness |
| | | n = 76 | |
| | | Age *(yr)*, mean (SD) = 53.6 (10.9) | |
| | | Sex, number male (%) = 63 (83) | |
| | | Sample included people with spasticity = yes | |
| Rand & Eng (2015) [142] | ARAT | Diagnosis = Stroke | Construct validity |
| | | Time since diagnosis *(mo)* = less than 6 | |
| | | n = 32 | |
| | | Age *(yr)*, mean (SD) = 58.1 (12.4) | |
| | | Sex, number male (%) = 25 (78) | |
| | | Sample included people with spasticity = not reported | |

(*Continued*)

**Table 2.** (Continued)

| Studies included | Measurement tool | Summary of study participants | Psychometric property tested |
|---|---|---|---|
| Riazi et al., (2003) [143] | SF-36 | Diagnosis = Multiple Sclerosis | Construct validity |
| | | Time since diagnosis *(mo)* = greater than 6 | |
| | | n = 638 | |
| | | Age *(yr)*, range = 20 - >60 | |
| | | Sex, number male (%) = 219 (35) | |
| | | Sample included people with spasticity = not reported | |
| Rigby et al., (2009) [144] | OHS | Diagnosis = Stroke | Construct validity |
| | | Time since diagnosis *(mo)* = less than 6 | |
| | | n = 104 | |
| | | Age *(yr)*, mean (SD) = 72.0 (13) | |
| | | Sex, number male (%) = 43 (41) | |
| | | Sample included people with spasticity = not reported | |
| Robinson et al (2009) [145] | SF-36 | Diagnosis = MS | Construct validity |
| | | Time since diagnosis *(mo)* = greater than 6 | Interpretability |
| | | n = 249 | |
| | | Age *(yr)*, mean (range) = 39 (10.5) | |
| | | Sex, number male (%) = 75 (30) | |
| | | Sample included people with spasticity = not reported | |
| Sabari et al., (2005) [146] | UL-MAS | Diagnosis = Stroke | Structural validity |
| | | Time since diagnosis *(mo)* = less than 6 (83%) | Interpretability |
| | | n = 100 | |
| | | Age *(yr)*, mean (range) = 54 (18–94) | |
| | | Sex, number male (%) = 67 (67) | |
| | | Sample included people with spasticity = not reported | |
| Sackley (1990) [147] | RMA | Diagnosis = Stroke | Construct validity |
| | RMA-UL | Time since diagnosis *(mo)* = less than 6 | |
| | | n = 52 (R hemiparesis), 38 (L hemiparesis) | |
| | | Age *(yr)*, mean (SD) = 63.4 (11.4) (R hemiparesis), | |
| | | 63.2 (11.9) (L hemiparesis) | |
| | | Sex, number male (%) = 33 (64) (R hemiparesis), | |
| | | 23 (61) (L hemiparesis) | |
| | | Sample included people with spasticity = not reported | |
| Salter et al., (2008) [148] | SF-36 | Diagnosis = Stroke | Content validity |
| | EQ-5D | Time since diagnosis *(mo)* = not reported | |
| | SIS | n = not reported | |
| | | Age *(yr)*, mean (SD) = not reported | |
| | | Sex, number male (%) = not reported | |
| | | Sample included people with spasticity = not reported | |
| Sarker et al., (2012) [149] | BI (C&W) | Diagnosis = Stroke | Construct validity |
| | | Time since diagnosis *(mo)* = greater than 6 | Interpretability |
| | | n = 238 | |
| | | Age *(yr)*, mean (SD) = 68.6 (14.2) | |
| | | Sex number male (%) = 124 (52) | |
| | | Sample included people with spasticity = not reported | |

(*Continued*)

**Table 2.** (Continued)

| Studies included | Measurement tool | Summary of study participants | Psychometric property tested |
|---|---|---|---|
| Schwid et al., (2002) [150] | NHPT | Diagnosis = Multiple Sclerosis | Measurement error |
| | | Time since diagnosis = unknown | |
| | | n = 27 | |
| | | Age *(yr)*, mean (SD) = 51.9 (9.0) | |
| | | Sex, number male (%) = 16 (79) | |
| | | Sample included people with spasticity = not reported | |
| Sharrack et al., (1999) [151] | BI (C&W) | Diagnosis = Multiple Sclerosis | Internal consistency |
| | FIM | Time since diagnosis *(mo)* = greater than 6 | Reliability |
| | | n = 25–64 | Structural validity |
| | | Age *(yr)*, median (range) = 40 (42.1–77.6) | Construct validity |
| | | Sex, number male (%) = 22 (34) | Responsiveness |
| | | Sample included people with spasticity = not reported | |
| Simon et al., (2008) [152] | OHS | Diagnosis = Stroke | Construct validity |
| | | Time since diagnosis *(mo)* = less than 6 | |
| | | n = 53 | |
| | | Age *(yr)*, mean (SD) = 65.6 (12.1) | |
| | | Sex, number male (%) = 14 (28) | |
| | | Sample included people with spasticity = not reported | |
| Stineman et al., (1996) [153] | FIM | Diagnosis = mixed (Stroke = 26, 183, Traumatic Brain Injury = 3, 214) | Internal consistency |
| | | Time since diagnosis *(mo)* = less than 6 | Structural validity |
| | | n = 29 397 | |
| | | Age *(yr)*, mean range = 41.6–71.3 | |
| | | Sex, number male (%) = not reported | |
| | | Sample included people with spasticity = not reported | |
| Stone et al., (1993) [154] | MI | Diagnosis = Stroke | Construct validity |
| | | Time since diagnosis *(mo)* = less than 6 | |
| | | n = 84 | |
| | | Age *(yr)*, mean (SD) = 72.37 (12.11) | |
| | | Sex, number male (%) = not given | |
| | | Sample included people with spasticity = not reported | |
| Sturm et al., (2002) [155] | AQoL | Diagnosis = Stroke | Construct validity |
| | | Time since diagnosis *(mo)* = less than 6 | Interpretability |
| | | n = 93 | |
| | | Age *(yr)*, mean (range) = 72 (28–89) | |
| | | Sex, number male (%) = 42 (45) | |
| | | Sample included people with spasticity = not reported | |
| Turner-Stokes et al., (2010) [156] | GAS | Diagnosis = Stroke | Construct validity |
| | | Time since diagnosis *(mo)* = greater than 6 | |
| | | n = 90 | |
| | | Age *(yr)*, mean (SD) = 54.5 (13.2) | |
| | | Sex, number male (%) = 54 (60) | |
| | | Sample included people with spasticity = yes | |

(*Continued*)

**Table 2.** (Continued)

| Studies included | Measurement tool | Summary of study participants | Psychometric property tested |
|---|---|---|---|
| Uswatte & Taub (2005) [157] | MAL | Diagnosis = not reported | Content validity |
| | | Time since diagnosis (*mo*) = not reported | |
| | | n = not reported | |
| | | Age (*yr*), mean (SD) = not reported | |
| | | Sex number male (%) = not reported | |
| | | Sample included people with spasticity = not reported | |
| Uswatte et al., (2006) [158] | MAL | Diagnosis = Stroke | Internal consistency |
| | MAL-28 | Time since diagnosis (*mo*) = greater than 6 | Reliability |
| | | n = 222 | Content validity |
| | | Age (*yr*), mean (SD) = 62.2 (13.0) | Structural validity |
| | | Sex number male (%) = 142 (64) | Interpretability |
| | | Sample included people with spasticity = not reported | |
| Van der Putten et al., (1999) [159] | BI (C&W) | Diagnosis = Mixed (Stroke n = 82, Multiple Sclerosis n = 201) | Responsiveness |
| | FIM | Time since diagnosis (*mo*) = less than 6 | Interpretability |
| | | n = 283 | |
| | | Age (*yr*), mean (SD) = 52 (16.9) (Stroke), | |
| | | 45 (11.2) (Multiple Sclerosis) | |
| | | Sex number male (%) = 238 (84) | |
| | | Sample included people with spasticity = not reported | |
| Van Straten et al (1997) [160] | SA-SIP30 | Diagnosis = Stroke | Content validity |
| | | Time since diagnosis (*mo*) = less than 6 | |
| | | n = 319 | |
| | | Age (*yr*), mean (SD) = 69 (12.6) | |
| | | Sex number male (%) = 175 (55) | |
| | | Sample included people with spasticity = not reported | |
| Vickrey et al., (1997) [161] | SF-36 | Diagnosis = Multiple Sclerosis | Internal consistency |
| | | Time since diagnosis (*mo*) = greater than 6 | Reliability |
| | | n = 171 (internal consistency, hypothesis testing), | Construct validity |
| | | n = 84 (reliability) | |
| | | Age (*yr*), mean (range) = 45 (20–67) | |
| | | Sex, number male (%) = 123 (72) | |
| | | Sample included people with spasticity = not reported | |
| Vickrey et al., (1995) [162] | SF-36 | Diagnosis = Multiple Sclerosis | Construct validity |
| | | Time since diagnosis (*mo*) = greater than 6 | |
| | | n = 179 | |
| | | Age (*yr*), mean (range) = 45 (20–67) | |
| | | Sex, number male (%) = 129 (72) | |
| | | Sample included people with spasticity = not reported | |
| Wade & Hewer (1987) [163] | BI (C&W) | Diagnosis = Stroke | Structural validity |
| | MI | Time since diagnosis (*mo*) = less than 6 | Construct validity |
| | | n = 976 | |
| | | Age (*yr*), mean (SD) = not given | |
| | | Sex, number male (%) = not given | |
| | | Sample included people with spasticity = not reported | |

(*Continued*)

**Table 2.** (Continued)

| Studies included | Measurement tool | Summary of study participants | Psychometric property tested |
|---|---|---|---|
| Wallace et al., (2002) [164] | BI | Diagnosis = Stroke | Responsiveness |
| | | Time since diagnosis *(mo)* = less than 6 | |
| | | n = 372 | |
| | | Age *(yr)*, mean (SD) = 69.7 (11.6) | |
| | | Sex number male (%) = 177 (48) | |
| | | Sample included people with spasticity = not reported | |
| Ware & Sherbourne (1992) [165] | SF-36 | Diagnosis = not reported | Content validity |
| | | Time since diagnosis *(mo)* = not reported | |
| | | n = not reported | |
| | | Age *(yr)*, mean (SD) = not reported | |
| | | Sex number male (%) = not reported | |
| | | Sample included people with spasticity = not reported | |
| Wellwood et al., (1995) [166] | BI | Diagnosis = Stroke | Construct validity |
| | | Time since diagnosis *(mo)* = greater than 6 | Interpretability |
| | | n = 152 | |
| | | Age *(yr)*, mean (SD) = 73 (13.4) | |
| | | Sex number male (%) = 68 (45) | |
| | | Sample included people with spasticity = not reported | |
| Wilkinson et al., (1997) [167] | BI (C&W) | Diagnosis = Stroke | Construct validity |
| | | Time since diagnosis *(mo)* = greater than 6 | Interpretability |
| | | n = 106 | |
| | | Age *(yr)*, median (range) = 71 (34–79) | |
| | | Sex number male (%) = 57 (54) | |
| | | Sample included people with spasticity = not reported | |
| Williams et al., (1999) [168] | SF-36 | Diagnosis = Stroke | Construct validity |
| | | Time since diagnosis *(mo)* = less than 6 | |
| | | n = 71 | |
| | | Age *(yr)*, mean (SD) = 61 (13) | |
| | | Sex, number male (%) = 45 (63) | |
| | | Sample included people with spasticity = not reported | |
| Williams (1990) [169] | EQ-5D | Diagnosis = not reported | Content validity |
| | | Time since diagnosis *(mo)* = not reported | |
| | | n = not reported | |
| | | Age *(yr)*, mean (SD) = not reported | |
| | | Sex, number male (%) = not reported | |
| | | Sample included people with spasticity = not reported | |
| Wolf & Koster et al., (2013) [170] | SIS | Diagnosis = Stroke | Construct validity |
| | | Time since diagnosis *(mo)* = greater than 6 | |
| | | n = 96 | |
| | | Age *(yr)*, median (range) = Grp 1 64.2 (13.4), Grp 2 60.5 (12.8) | |
| | | Sex, number male (%) = Grp 1 28 (52), Grp 2 31 (55) | |
| | | Sample included people with spasticity = not reported | |

**Table 2.** (*Continued*)

| Studies included | Measurement tool | Summary of study participants | Psychometric property tested |
|---|---|---|---|
| Xie et al., (2006) [171] | EQ-5D | Diagnosis = Stroke | Construct validity |
| | | Time since diagnosis (*mo*) = not reported | |
| | | n = 1040 | |
| | | Age (*yr*) = ≥18 | |
| | | Sex, number male (%) = 447 (43.9) | |
| | | Sample included people with spasticity = not reported | |
| Yozbatiran et al., (2008) [172] | ARAT | Diagnosis = Stroke | Reliability |
| | | Time since diagnosis (*mo*) = greater than 6 | Construct validity |
| | | n = 12 (validity) n = 9 (interrater reliability) n = 8 (intra rater) | |
| | | Age (*yr*), mean (SD) = 61.0 (15.0) | |
| | | Sex, number male (%) = 6 (50) | |
| | | Sample included people with spasticity = not reported | |
| | | Rater characteristics | |
| | | Rater n = 2 Clinical experience (*yr*) = 8 | |
| | | Observations n = 58 | |

RMA = Rivermead Motor Assessment, RMA-UL = Rivermead Motor Assessment–Upper Limb, BI (C&W) = Barthel Index Collin & Wade version, EQ-5D = EuroQol -5 dimension, SIS = Stroke Impact Scale, SF-36 = Medical Outcome Study 36-Item Short-Form Health Survey, ArmA = Arm Activity Measure, ARAT = Action Research Arm Test, NHPT = Nine Hole Peg Test, MI = Motricity Index, GAS = Goal Attainment Scale, DAS = Disability Assessment Scale, FIM = Functional Independence Measure, UL-MAS = Upper Limb–Motor Assessment Scale, CMSA = Chedoke-McMaster Stroke Assessment, SA-SIP30 = Stroke-Adapted Version of the Sickness Impact Profile, MAL = Motor Activity Log, BI = Barthel Index, AQoL = Assessment of Quality of Life, mFAT = modified Frenchay Arm Test, OHS = Oxford Handicap Scale, MAL-28 = Motor Activity log– 28.

evidence for each measurement tools is presented in Table 3. Due to the volume of data, summaries of individual study results and psychometric properties tested are tabulated within S2 and S3 Tables. The following summarizes the appraisal of each tool. *These have been placed in alphabetical order.*

**Action Research Arm Test.** The Action Research Arm Test (ARAT) [173] is an obervational performance test that evaluates a person's ability to use their upper limb to handle objects using grasp, grip, pinch and gross motor movements. Twelve studies evaluated the psychometric properties of the ARAT [38, 45, 57, 63, 75, 113, 114, 120, 129, 140, 142, 172], four of those studies specifically identified participants with upper limb spasticity [38, 45, 75, 114]. The majority of studies included participants post-stroke with a single study including a mixed sample, post-stroke and TBI [120].

*Content validity.* The Upper Extremity Function Test (UEFT) [174] was modified by Lyle [173] to produce the ARAT. No further content validity studies were identified. The ARAT was found to have sufficient relevance, but indeterminant ratings for comprehensiveness and comprehensibility and no participants were interviewed regarding those properties.

*Results for whole sample.* Research supports hierarchical ordering of items [173] and reliability within (ICC = 0.99) and between raters (ICC 0.99) [172]. The ARAT was found to correlate highly with other like-tests of activity and dexterity (r = 0.65–0.95) [57, 63, 129, 140, 142, 172] and weak to moderately with the Functional Independence Measure (FIM), a more global measure of function (r = 0.47) [140]. ARAT scores were not, however, a predictor of overall quality of life [129]. The ARAT was found to be responsive over time in acute as well as chronic stroke and TBI samples [38, 57, 114, 140]. ARAT was found to be equally sensitive to

**Table 3. Synthesis of evidence.**

| Measurement tool | Sample | Content validity | Structural validity | Internal consistency | Cross cultural validity | Reliability Inter | Reliability Intra | Reliability Retest | Measurement error | Construct validity | Responsiveness |
|---|---|---|---|---|---|---|---|---|---|---|---|
| ARAT | *Spasticity* | | | | | | | | | *Moderate* | *Low* |
| | *n = 4* | | | | | | | | | *- (13/21)* | *+ (4/4)* |
| | Whole sample | Very Low | Very Low | | | Very Low | Very Low | | | Moderate | Moderate |
| | n = 12 | | + | | | + | + | | | - (19/30) | + (6/6) |
| ArmA | *Spasticity* | *High* | *High* | *Moderate* | | | | *Low* | | *Very Low* | *Moderate* |
| | *n = 5* | | + | + | | | | + | | + | + (4/4) |
| | Whole sample | High | High | Moderate | | | | Low | | Very Low | Moderate |
| | n = 5 | | + | + | | | | + | | + | + (4/4) |
| AQoL | *Spasticity* | | | | | | | | | | |
| | *n = 0* | | | | | | | | | | |
| | Whole sample | Very Low | | | | | | | | High | |
| | n = 3 | | | | | | | | | + (3/3) | |
| BI | *Spasticity* | | | | | | | | | | |
| | *n = 0* | | | | | | | | | | |
| | Whole sample | Very Low | | | | | | | | High | Very Low |
| | n = 6 | | | | | | | | | + (5/6) | - (0/1) |
| BI (C&W) | *Spasticity* | | | | | | | | | | |
| | *n = 0* | | | | | | | | | | |
| | Whole sample | Very Low | Low | | | Very Low | | Very Low | Very Low | Moderate | Low |
| | n = 9 | | + | | | ? | | ? | + | + | - (2/3) |
| CMSA | *Spasticity* | | | | | | | | | | |
| | *n = 0* | | | | | | | | | | |
| | Whole sample | Very Low | | | | Moderate + | Moderate | Low | | Moderate | Very Low |
| | n = 4 | | | | | Low +* | + | + | | + (5/6) | + (1/1) |
| DAS | *Spasticity* | *Very Low* | | | | *Low* | *Low* | | | *Moderate* | |
| | *n = 2* | | | | | ? | - | | | + (2/2) | |
| | Whole sample | Very Low | | | | Low | Low | | | Moderate | |
| | n = 2 | | | | | ? | - | | | + (2/2) | |
| EQ-5D | *Spasticity* | | | | | | | | | *High* | |
| | *n = 2* | | | | | | | | | + (3/3) | |
| | Whole sample | Moderate | | | | | | Moderate +^ | | Moderate | Low |
| | n = 19 | ? | | | | | | Very Low - ^^ | | + (24/34) | - (11/15) |
| FAT | *Spasticity* | | | | | | | | | | |
| | *n = 0* | | | | | | | | | | |
| | Whole sample | | | | | | | | | | |
| | n = 0 | | | | | | | | | | |

*(Continued)*

**Table 3.** (Continued)

| Measurement tool | Sample | Content validity | Structural validity | Internal consistency | Cross cultural validity | Reliability | | | Measurement error | Construct validity | Responsiveness |
|---|---|---|---|---|---|---|---|---|---|---|---|
| | | | | | | Inter | Intra | Retest | | | |
| mFAT | *Spasticity* | | | | | | | | | | |
| | *n = 0* | | | | | | | | | | |
| | Whole sample n = 2 | | | | | Very Low ? | | Very Low ? | | Very Low - (0/1) | |
| FIM | *Spasticity* | | | | | | | | | *Moderate* + (1/1) | *Very Low* + (1/1) |
| | *n = 1* | | | | | | | | | | |
| | Whole sample | Very Low | High | High | | Moderate | Low | | | High | Moderate |
| | n = 20 | | + | + | | + | + | | | + (23/29) | - (5/7) |
| Global Ax | *Spasticity* | | | | | | | | | | |
| | *n = 0* | | | | | | | | | | |
| | Whole sample n = 0 | | | | | | | | | | |
| GAS | *Spasticity* | | | | | | | | | *Very Low* – (3/7) | |
| | *n = 1* | | | | | | | | | | |
| | Whole sample n = 9 | | | | | Low - | | | Low ? | Moderate – (14/23) | Low + (4/4) |
| GAS-10pt | *Spasticity* | | | | | | | | | | |
| | *n = 0* | | | | | | | | | | |
| | Whole sample n = 0 | | | | | | | | | | |
| Klein-Bell | *Spasticity* | | | | | | | | | | |
| | *n = 0* | | | | | | | | | | |
| | Whole sample n = 0 | | | | | | | | | | |
| LASIS | *Spasticity* | | | | | | | | | | |
| | *n = 0* | | | | | | | | | | |
| | Whole sample n = 0 | | | | | | | | | | |
| MAL | *Spasticity* | | | | | | | | | *Low* - (3/7) | |
| | *n = 1* | | | | | | | | | | |
| | Whole sample n = 5 | Very Low | Very Low ? | | | | | | Low ? | Moderate - (4/9) | |
| MAL-5 | *Spasticity* | | | | | | | | | | |
| | *n = 0* | | | | | | | | | | |
| | Whole sample n = 0 | | | | | | | | | | |

(*Continued*)

**Table 3.** (Continued)

| Measurement tool | Sample | Content validity | Structural validity | Internal consistency | Cross cultural validity | Reliability | | | Measurement error | Construct validity | Responsiveness |
|---|---|---|---|---|---|---|---|---|---|---|---|
| | | | | | | Inter | Intra | Retest | | | |
| MAL-28 | *Spasticity* | | | | | | | | | | |
| | *n = 0* | | | | | | | | | | |
| | Whole sample | Very Low | Very Low | Very Low | | | | Moderate +^ | | Very Low + (3/4)^ | |
| | n = 1 | | ? | +** | | | | Low -^^ | | Very Low– (2/4)^^ | |
| MI | *Spasticity* | | | | | | | | | | |
| | *n = 0* | | | | | | | | | | |
| | Whole sample n = 6 | Very Low | | Very Low ? | | Very Low ? | | | | | Moderate - (4/6) | Very Low - (0/1) |
| NHPT | *Spasticity* | | | | | | | | | | *Very Low* |
| | *n = 1* | | | | | | | | | | *- (3/5)* | *+ (2/2)* |
| | Whole sample | | | | | Very Low | | Very Low | Very Low | Moderate | Low |
| | n = 10 | | | | | ? | | ? | + | - (21/32) | + (3/3) |
| OHS | *Spasticity* | | | | | | | | | | |
| | *n = 0* | | | | | | | | | | |
| | Whole sample n = 2 | | | | | | | | | | Low - (2/3) | |
| PDS / CBS | *Spasticity* | | | | | | | | | | |
| | *n = 0* | | | | | | | | | | |
| | Whole sample n = 0 | | | | | | | | | | | |
| RMA | *Spasticity* | | | | | | | | | | |
| | *n = 0* | | | | | | | | | | |
| | Whole sample n = 5 | Very Low | Very Low - | | | | | | | | High + (2/2) | |
| RMA–UL | *Spasticity* | | | | | | | | | | |
| | *n = 0* | | | | | | | | | | |
| | Whole sample n = 6 | Very Low | Very Low +, - ^^^ | | | | | | | | High + (3/4) | |
| SF-36 | *Spasticity* | | | | | | | | | | |
| | *n = 0* | | | | | | | | | | |
| | Whole sample n = 24 | Very Low | Moderate ? | High + | | | | Moderate +^ Low -^^ | | Moderate– (25/44) | Very Low–(0/4) |
| SA-SIP | *Spasticity* | | | | | | | | | | *Moderate* |
| | *n = 1* | | | | | | | | | | *+ (1/1)* |
| | Whole sample | Moderate | | | | | | | | | High |
| | n = 4 | | | | | | | | | | + (3/3) |

*(Continued)*

**Table 3.** (Continued)

| Measurement tool | Sample | Content validity | Structural validity | Internal consistency | Cross cultural validity | Reliability | | | Measurement error | Construct validity | Responsiveness |
|---|---|---|---|---|---|---|---|---|---|---|---|
| | | | | | | Inter | Intra | Retest | | | |
| SIS | *Spasticity* | | | | | | | | | | |
| | *n = 0* | | | | | | | | | | |
| | Whole sample n = 10 | Moderate | High | Moderate | | Low | | Low | | High | |
| | | + | + | + | | ? | | + | | + (18/19) | |
| UL-MAS | *Spasticity* | | | | | | | | | | |
| | *n = 0* | | | | | | | | | | |
| | Whole sample n = 10 | Very Low | Moderate | Moderate | | Low | Low | | | Moderate | |
| | | | + | +** | | ? | ? | | | - (3/8) | |

*High* = Very confident that the true measurement property lies close to that of the estimate of the measurement property. *Moderate* = Moderate confidence in the measurement property estimate. *Low* = Limited confidence in the measurement property estimate. *Very low* = Little confidence in the measurement property estimate, full definition of ratings reported in [9]. + = sufficient,—insufficient,? indeterminant [9].

*Moderate + Impairment Inventory, Low + Activity Inventory

**Internal consistency evidence strength cannot exceed structural validity as per COSMIN guidelines and has been reduced accordingly.

^Patients reports

^^ proxy reports

^^^ '+' acute sample, '-' subacute sample.

ARAT = Action Research Arm Test, ArmA = Arm Activity Measure, AQoL = Assessment of Quality of Life, BI = Barthel Index, BI (C&W) = Barthel Index—Collin & Wade version, CMSA = Chedoke-McMaster Stroke Assessment, DAS = Disability Assessment Scale, EQ-5D = EuroQol– 5 dimension, FAT = Frenchay Arm Test, mFAT = modified Frenchay Arm Test, FIM = Functional Independence Measure, GAS = Goal Attainment Scale, GAS– 10pt = Goal Attainment Scale– 10 point, Global Ax = Global Assessment Scale, KleinBell ADL = Klein-Bell Activities of Daily Living scale, LASIS = Leeds Adult Spasticity Impact Scale, SF-36 = Medical Outcome Study 36-Item Short-Form Health Survey, MAL = Motor Activity Log, MAL-5 = Motor Activity Log—5, MAL-28 = Motor Activity Log—28, MI = Motricity Index, NHPT = Nine Hole Peg Test, OHS = Oxford Handicap Scale, PDS/CBS = Patient Disability Scale / Carer Burden Scale, RMA = Rivermead Motor Assessment, RMA-UL = Rivermead Motor Assessment—Upper Limb, SA-SIP = Stroke-Adapted Version of the Sickness Impact Profile, SIS = Stroke Impact Scale, UL MAS = Upper Limb Motor Assessment Scale.

change as like measures when used with participants less than 6 months post-stroke [57, 140]. Mixed results have been reported with respect to ceiling effect in stroke populations [63, 75] and there is one study which has reported a minimal, clinically important change of 12 points (dominant) and 17 (non-dominant) [113].

*Results pertaining to sample with upper limb spasticity*. The ARAT correlated strongly with like measures of activity and dexterity (r = 0.69–0.95) [38] and less with a global measure of function (Functional Independence Measure (FIM) r = 0.2–0.6) [114] and impairments, including grip and pinch strength, spasticity and AROM (r = - 0.28–0.86) [38, 45, 114]. The ARAT was moderate to highly responsive to capture change in participants less than 6 months post-stroke (ES = 0.55–1.018) [38, 114], being as equally responsive as like measures (NHPT and Jebsen-Taylor test of hand function), more responsive than measures of impairment (pinch and grip strength), but less responsive than the SIS-Hand (ES = 0.55–1.018) [38]. Neither a floor nor ceiling effects were found in a sample of participants greater than 6 months post-stroke [75].

**Arm Activity measure.** The Arm Activity measure (ArmA) is a 20-item self-report tool which includes 7 passive and 13 active items to capture real arm activity in neurological populations [33]. Five studies [30–34] evaluated the psychometric properties of the ArmA, the

majority of studies included a mixed sample including participants post-stroke, TBI and MS. All included studies specifically identified participants with upper limb spasticity.

*Content validity*. The ArmA was developed based on goal analysis, systematic literature review and a modified Delphi survey which demonstrated relevance, comprehensiveness and comprehensibility [30, 33].

*Results pertaining to sample with upper limb spasticity*. The ArmA subscales demonstrated internal consistency (passive subscale α = 0.85, active subscale α = 0.96) and retest reliability (quadratic weight kappa 0.90 (CI 0.68–1.12), active subscale 0.93 (CI 0.71–1.15)) in a sample with upper limb spasticity [34]. The ArmA demonstrated convergent and divergent validity with passive and active items of the Leeds Adult Spasticity Scale (LASIS) and Disabilities of Arm Shoulder and Hand (DASH) (convergent: Rho 0.48; p = 0.01 to 0.63; p = 0.01; divergent: Rho 0.02; p = 0.9 to 0.23; p = 0.078) [34] and was found to be responsive [32, 34]. Preliminary analysis suggests clinically meaningful change is indicated by 2.5 or 3 point improvement (passive subscale) and 1.1 or 2.5 point improvement (active subscale) [34]. The ArmA active function subscale suffered a ceiling effect (37%), however no floor effect was observed for either subscale [34].

**Assessment of Quality of Life.** The Assessment of Quality of Life (AQoL) is a generic HRQoL measure that assesses independent living, social relationships, physical senses, psychological wellbeing and illness [90]. Three studies evaluated the psychometric properties of the AQoL, one included participants greater than 6 months post TBI [89] and two less than 6 months post-stroke [90, 154]. Neither study specifically identified participants with upper limb spasticity.

*Content validity*. Development research underpinning the AQoL [90] demonstrated sufficient relevance, but indeterminant ratings for comprehensiveness and comprehensibility. No other content validity studies conducted in a neurological sample were identified.

*Results for whole sample*. The AQoL discriminated between participants with and without TBI (effect size (ES) = 0.80), with participants post TBI scoring 2.0 utilities lower than participants without [89]. The AQoL correlated more strongly with measures of handicap (London Handicap Scale (LHS) r = 0.83) than disability (Barthel Index (BI) r = 0.77) or impairment (National Institute of Health Stroke Scale (NIHSS) r = -0.69) in the first 6 months post-stroke and was a significant predictors of death or institutionalization at 12 months [155]. No floor or ceiling effects (1–2%) were found in a stroke population [155].

**Barthel Index.** The Barthel Index (BI) was initially developed to score the abilities of participants to care for themselves [123]. The BI evaluates 10 activity areas, with a maximum score of 100 indicating independence in all included areas. Six studies evaluated the psychometric properties of the BI [28, 72, 111, 123, 164, 166]. Five studies were completed with participants post-stroke, 4 included participants less than 6 months post-stroke [28, 72, 111, 164], 1 greater than 6 months post-stroke [166] and 1 discussed tool development with a non-specific sample [123]. No included studies specifically identified participants with upper limb spasticity.

*Content validity*. No research on the development of the BI was located.

*Results for whole sample*. The BI correlated moderately with measures of upper limb function (Fugl-Meyer Rho = 0.60) (Functional Test for the Hemiplegic/Paretic Upper Limb Rho = 0.61) [72] and global measures function (FIM $r_s$ = 0.95, p<0.0001; Modified Rankin Scale (MRS) $r_s$ = 0.89, p<0.0001; Office of Population Censuses and Surveys (OPCS) disability instrument r = 0.73, p<0.001) [111, 166]. The BI was equally responsive to change within the first three months post-stroke as like global measures (FIM) [164] and a measure of motor function (Fugl-Meyer Test) [72], however determined responsiveness was low. Evidence of a ceiling effect was found in a sample greater than 6 months post-stroke [166].

**Barthel Index (Collin & Wade).** The Barthel Activities of Daily Living Index (BI C&W) [49] is a modification of the original BI measurement tool, with all 10 areas of activity included but is scored in increments of 1 rather than 5 as per the original BI [123]. Nine studies evaluated the psychometric properties of the BI(C&W) [35, 49, 56, 83, 97, 149, 159, 163, 167], 6 studies included participants post-stroke [35, 56, 83, 149, 163, 167] and 3 included mixed samples (stroke, MS, TBI) [49, 97, 159]. No studies specifically identified participants with upper limb spasticity.

*Content validity.* No information presenting the methodology used to revise the original BI was found, only justification from revised test authors who felt the original five-point incremental scoring was misleading in accuracy [49].

*Results for whole sample.* Research supports use of a summed BI(C&W) score due to a single factor (68% of variance) underlying the scale [163]. While the hierarchical nature of the BI (C&W) was supported by Wade and Hewer [163], Barer and Murphy [35] reported a failure to meet Guttman scaling criteria. Test-retest reliability results appear mixed, with high agreement (75%) between scores but variations in kappa (-0.99 to 0.81) [83]. Inter-rater reliability between self-report, family, nursing staff and skilled observers was acceptable (agreement within 2 points or less for 72% of participants) [49]. The BI(C&W) was strongly associated with measures of upper limb activity (r = 0.729–0.826) (Motricity Index Upper Limb (MI UL) and Motricity Index (MI) total, Frenchay Activity Index (FAI)), complex daily activities (r $\geq$ 0.80), and disability ($r_s$ = 0.726–0.80) (London Handicap Scale, Modified Rankin Scale (MRS)), and less with measures of psychological wellbeing and impairments (depression, anxiety, pain) (r = 0.2–0.423) [56, 149, 163, 167]. Research suggests that BI(C&W) is at least equally responsive to FIM [97, 159]. However, BI(C&W) suffered from floor and ceiling effects across the acute through to community continuum in a mixed neurorehabilitation sample [97, 149, 159, 167].

**Chedoke-McMaster Stroke Assessment.** The Chedoke-McMaster Stroke assessment (CMSA) is comprised of two parts; the impairment inventory and the activity inventory (formerly known as the disability inventory) [81]. The CMSA impairment inventory classifies participants into subgroups based on the stages of motor recovery, while the CMSA activity inventory provides a measure of activity performance. Four studies evaluated the psychometric properties of the CMSA, two included participants less than 6 months post-stroke[54, 81], two did not report on the length of time post-stroke for participants [80, 128] and no study specifically identified participants with upper limb spasticity.

*Content validity.* Evidence located for the development of the CMSA [80, 128], did not indicate participants were consulted on the comprehensiveness or comprehensibility of included items. Relevance of items for the intended purpose of assessment of stroke clients within rehabilitation setting was sufficient, however further content validity studies were not identified.

*Results for whole sample.* Evidence supports the reliability of the CMSA; inter-rater (ICC 0.88 (95% CI 0.76–0.94) to 0.99 (95%CI 0.98–1.00)), intra-rater (ICC 0.93 (95% CI 0.85–0.96) to 0.98 (95% CI 0.95–0.99)), test retest (ICC 0.98 (95% CI 0.95–0.99)) [81]. Consistent with the definition of the CMSA, strong correlations with both subscales and total scores for like measures of upper limb activity performance (Fugl-Meyer r = 0.95, p<0.001) and global measures of function (FIM r = 0.79, p<0.05) were demonstrated [81]. The predictive validity through use of the Gowland's predictive equations, however, were not supported due to large error associated with the predicted value [54]. The CMSA was found to be more responsive than the FIM when used with participants less than 6 months post-stroke [81].

**Disability Assessment Scale.** The Disability Assessment Scale (DAS) is a brief measure of functional disability [42]. Two studies were included, both identified participants with upper limb spasticity [42, 58].

*Content validity.* Brashear and colleagues [42] reported the development of the DAS to fill the identified gap within the evaluation of functional impairment commonly seen in participants with post-stroke upper limb spasticity (i.e. dressing, hygiene, limb position, pain). No additional research underpinning measurement tool development was reported.

*Results pertaining to sample with upper limb spasticity identified.* Good to excellent intra-rater reliability (78% of evaluations weighted kappa ≥ .4) and good inter-rater reliability (Kendall W 0.49 (95% CI 0.30–1.00, p < .001) to 0.77 (95% CI 0.37–1.00, p < .001) was reported when used by professionals (neurologists, physiatrists, occupational therapists and physical therapists) with a mean of 6 years clinical experience [42]. Greater DAS scores were found to be associated with Stroke-Adapted Version of the Sickness Impact Scale (SA-SIP) scores (P < .05), reduced quality of life and caregiver burden (P < .05) [58, 175].

**EuroQol-5 dimension.** The EuroQol-5 dimension (EQ-5D) is a generic measure of health-related quality of life [73, 78, 169]. Nineteen studies evaluated the psychometric properties of the EQ-5D, including participants with MS (n = 6), [73, 106–108, 127, 131] a mixed neurological sample (n = 1) [27] and post-stroke (n = 12) [36, 37, 58, 60–62, 78, 135, 136, 148, 171]. Two studies specifically identified participants with upper limb spasticity [58, 78].

*Content validity.* During the development of the EQ-5D there is no evidence that participants were consulted on the comprehensiveness or comprehensibility of included items. Relevance of items for the intended purpose was sufficient [108]. The EQ-5D contains 6 of 9 recommended dimensions for patient-based, health related quality of life measures and is less comprehensive than the Stroke Impact Scale (SIS) [148].

*Results for whole sample.* Test-retest reliability of the patient-reported EQ-5D was moderate to good for VAS and the mobility domain (ICC ≥0.70) [61, 73], test-retest reliability was lower in proxy-reported scores [61]. The EQ-5D correlated moderately with global measures of function such as the EDSS (r = -0.66) [73], but was less sensitive than disease-specific quality of life scales and the generic SF-36 when used with participants with MS [131]. A single study found a moderate inverse relationship between the EQ-5D and the Nine Hole Peg Test, a specific measure of upper limb use (r = -0.56) [73]. When used with participants post-stroke, the EQ-5D correlated with global measures of function including the SF-6D, a classification for describing health from a selection of SF-36 items (r = 0.77) [37] and the SF-36 (r = 0.57–0.63) [60]. Evidence of the discriminant ability was found between participants post-stroke and those who had not suffered a stroke [36, 171], between stroke type and severity [62], and between participants with and without spasticity [78]. The EQ-5D Index had the greatest change score when compared to like generic HRQoL measures less than 6 months post-stroke [136], was more responsive to changes in disability (MRS r = -0.36) and daily activities (BI r = 0.57) in comparison to the EQ-5D VAS [136]. Contrarily, neither the EQ-5D Index or VAS was responsive to change over a one year period post-stroke despite 23.8% of participants reporting improvement and 23.2% deterioration [135]. The EQ-5D did not demonstrate either floor and ceiling effects when used with acute participants post-stroke [136].

*Results pertaining to sample with upper limb spasticity identified.* The EQ-5D index scores were found to correlate with measures of disability (p < .002) and carer burden (p < .05) [58] and to distinguish between participants with and without upper limb spasticity post-stroke, with mean differences (-0.07, 95% CI -0.12 to -0.33) equivalent to the MCID established for the EQ-5D for other health conditions (MCID is yet to be established for post-stroke populations) [78].

**Modified Frenchay Arm Test.** The modified Frenchay Arm Test (mFAT), reduces the 25 clinical tests to 5 so as to measure arm function after stroke [94]. Two studies evaluated the psychometric properties of the mFAT [94]; no studies specifically identified participants with upper limb spasticity.

*Content validity.* No studies were identified providing information targeting measurement tool development and/or content validity.

*Results for whole sample.* There was evidence for the reliability of the mFAT (inter-rater (Rho = 0.75–0.99), test-retest (Rho = 0.68–0.90 and 0.83–0.99)) when administered to participants 18 months post-stroke [94]. The mFAT was found to be less sensitive than the NHPT in participants less than 6 months post-stroke with mild impairments [94]. Floor effects (30%) and ceiling effects (34%) were evident within acute stroke [94].

**Functional independence measure.** A total of 20 studies evaluated the psychometric properties, in participants post-stroke (n = 9) [44, 70, 82, 87, 92, 93, 111, 132, 134], TBI (n = 5) [50, 52, 53, 86, 91], MS (n = 2) [141, 151] and a mixed neurological sample (n = 3) [97, 153, 159]. One study specifically identified participants with upper limb spasticity in a sample with MS [141].

*Content validity.* The FIM was found to have sufficient relevance, but indeterminant ratings for comprehensiveness and comprehensibility during development, as nil information was located to determine if participants were interviewed regarding those properties [105].

*Results pertaining to whole sample.* A two factor structure was identified for the FIM by a number of researchers, with separate motor and cognitive domains accounting for 89.4 to 97.9% of variance [86, 92, 93, 151]. Evidence for internal consistency has been reported across a number of sample populations (complete FIM α = 0.94–0.98, FIM motor α = 0.93–0.97 and FIM cognitive α = 0.93–0.94 for stroke, MS, traumatic and non-traumatic samples [151, 153]). And between-rater reliability has been demonstrated for both the motor and cognitive domains of the FIM in acute stroke (ICC 0.96, 0.91) respectively [87] and with participants with MS (FIM total inter-rater ICC = 0.99, FIM total intra-rater ICC = 0.94) [151]. Predictive associations between FIM scores and length of stay, discharge destination, minutes of assistance and supervision required on discharge and return to driving were identified [44, 50, 52, 82, 91, 132, 134]. When used with participants with MS, FIM was found to be a valid measure of disability [141], strongly correlating with like global measures (BI r = 0.88), activity measures (Ambulation Index r = - 0.73) and moderate to strongly with specific activity measures including housework (r = 0.64, p<0.001), work (r = -0.59 p<0.001), independence (r = -0.44, p = 0.001), and disability r = -0.96, p< 0.001) [151]. The FIM total score was at best only moderately responsive to change in a neurorehabilitation sample (ES 0.52–0.72), but the FIM cognitive was not (ES = 0.35–0.43) [97]. In comparison to other measures, the FIM was found to be less responsive than the original BI, equally responsive to BI(C&W) in stroke and more responsive than EDSS in MS, yet still only weak to moderately responsive to change (FIM ES = 0.46, FIM SRM 0.53, EDSS 0.15) [141, 151, 159]. Evidence of floor and ceiling effects for FIM were also found [44, 151, 159].

*Results pertaining to sample with upper limb spasticity identified.* FIM scores correlated with a measures of disability (Kurtkze Expanded Disability Status Scale (EDSS) $r_s$ = -0.69) [141] and was found to be responsive when capturing change in participants with MS (SRM = 0.53) [141].

**Goal Attainment Scaling.** Goal Attainment Scaling (GAS) was first introduced by Kirusek and Sherman [176] and provides a structured approach to defining and measuring individualized patient centered and/or program based goals. A total of 9 studies evaluated the psychometric properties, in post-stroke (n = 2) [43, 156], MS (n = 1) [104], TBI (n = 3) [59, 120, 125] and mixed ABI (n = 3) samples [41, 115, 124]. Only one study met inclusion criteria that specifically identified participants with upper limb spasticity (in a sample greater than 6 months post-stroke) [156].

*Content validity.* Not assessed, as GAS identifies goal content particular to individual participants and programs (i.e. high face validity).

*Results for whole sample*. There were conflicting results in inter-rater reliability within a mixed neurological sample, while Joyce, Rockwood and Mate-Kole [102] report high reliability (r = 0.92, r = 0.94) between an individual rater familiar with GAS and the treating team, Bovend'Eerdt, Dawes, Izadi and Wade [41] found a fair level (ICC$_{A,k}$ 0.478) and low agreement (LOA -1.52 ± 25.54) between a therapist and masked assessor. When used with participants with MS, GAS change score correlated weakly with the BI (r$_s$ = -0.25) and FIM (r$_s$ = -0.6) [104]. In a sample of participants with ABI secondary to trauma and stroke, GAS also correlated strongly with global clinical impressions (r = 0.81) [104], weak to strongly with measures of daily activity, participation, disability, vocational outcome and quality of life (r = 0.34–0.81) but not with length of stay [102, 124, 125]. In the same sample, GAS at 2 months predicted final GAS scores at the completion of a rehabilitation program ranging from 7 to 42 weeks [125]. Ratings between participants and significant others agreed on 70% of occasions [59]. GAS was more responsive than the FIM and BI (ES 9.0 SRM: 2.4 t value 10.0 z value 1.4) in MS [104] and was responsive to patient centred outcomes and program change in a mixed neurological sample [115].

*Results pertaining to sample with upper limb spasticity*. GAS was found to have moderate correlations with self-reported benefit (rho = 0.46, p < .001), low correlations with quality of life (rho = 0.07, p = 0.52), disability (rho = 0.19, p = 0.08), carer burden (rho = 0.14, p = 0.26), measures of pain (rho = 0.03, p = 0.77), mood (rho = 0.06, p = 0.61) and spasticity (rho = 0.35, p = 0.001 [156].

**Medical Outcome Study 36-Item Short-Form Health Survey.** The Medical Outcome Study 36-Item Short-Form Health Survey (SF-36) is a global scale assessing eight health concepts [165, 177]. A total of 24 studies investigated the psychometric properties of the SF-36, 10 included participants with MS [76, 77, 95, 127, 130, 138, 143, 145, 161, 162], 10 post-stroke [29, 60, 61, 67, 85, 96, 122, 133, 148, 168], 3 post TBI [74, 84, 123] and 1 discussed tool development with nil specific sample [165]. No studies specifically identified participants with upper limb spasticity.

*Content validity*. The development of the SF-36 [165] did not appear to consult participants on the comprehensiveness or comprehensibility of included items [165]. Relevance of items for the intended purpose was sufficient. The SF-36 contains 6 of 9 recommended dimensions for patient-based, health related quality of life, less comprehensive than the SIS [148].

*Results for whole sample*. The SF-36 was found to have a two-factor structure; with the eight dimensions falling within the two constructs of physical and mental health [177]. Mixed results were found for the use of the domain scores, with scaling assumptions met in the TBI population [84] but only 6 of 8 scales meeting the scaling assumptions in stroke [96]. Evidence for internal consistency of the 8 dimensions, Cronbach alpha >0.70 in majority of studies [29, 61, 74, 76, 84, 161], however dimensions of vitality and general health did not meet this criteria (α = 0.68, α = 0.66–0.68) [85, 96]. Test-retest reliability varied; higher for patient reported scores (ICC = 0.30–0.81) than proxy reported scores (ICC = 0.25 to 0.76) [61, 130, 162]. Individual domains of the SF-36 correlated with like subscales of global measures (all r = ≥ 0.50) post-stroke (EQ-5D) [60] post TBI (Symptom Checklist, Health Problem List, Beck Depression Inventory) [74] and with participants with MS (LHS, FIM, general health questionnaire) [76]. Correlations, however, were not as strong as hypothesized between individual domains and like dimensions for the BI, CNS and FIM post stroke [85, 122] nor with the MSFC in a MS population (r = 0.16–0.51) [145]. The SF-36 physical and mental summary scores had weak to moderate correlations with participants rating of severity of symptoms (r = 0.38, r = 0.18) and quality of life (r = 0.47, r = 0.29) [127, 168]. The ability to discriminate between subgroups of participants with varying levels of function across post-stroke, TBI and MS populations was demonstrated [95, 138, 145, 161, 162]. The SF-36 was more responsive in the first three

months post-stroke [85] but less responsive in comparison to other tools measuring associated constructs in MS (ES = 0.01–0.30) [76]. SF-36 did not correlate with FIM change scores, suggesting the change captured within a HRQoL measure was not reflected in a global measure of activity [122]. There was evidence of significant floor and ceiling effects within MS [76, 77] and TBI [84], and varied reports post-stroke [60, 85, 96, 122, 133]. The minimal important clinical change varied across dimensions, reported to be 4–9 points within physical functioning, 6–8 within role physical, 6–7 social functioning and 6 points within the physical summary score [145].

**Motor Activity Log.** The Motor Activity Log (MAL) is a structured interview designed to capture use of the affected upper limb on two scales, Amount of Use (AOU) and Quality of Movement (QOM) [158]. Five studies evaluated the psychometric properties of MAL; all involved participants post-stroke [47, 63, 88, 157, 158], and one specifically identified participants with upper limb spasticity [88].

*Content validity*. The MAL was developed based on the non-use model to capture real-world arm function [157]. Item analysis suggests 2 items (put on makeup and write on paper) had greater than 20% missing data, with participants rating as not applicable, and had lower item-total correlations and reliability coefficients [158].

*Results for the whole sample*. The self-reported QOM scale correlated with performance based measures (ARAT r = 0.61, WMFT r = 0.65) with the AOU scale correlating less strongly with the WMFT r = 0.40 [63, 158]. The minimal detectable change was defined as 16.8% for the AOU and 15.3% for the QOM scales, but the minimal important change was not defined [47].

*Results pertaining to sample with upper limb spasticity*. The MAL correlated strongly with measures of activity (Chedoke Arm and Hand Activity Inventory (CAHAI) r = 0.82 p<0.01), weakly with measures of participation (Reintegration to Normal Living Index (RNL) r = 0.23 p<0.05) and of varying strengths (weak to moderate) with impairments, stronger than expected (spasticity r = -0.71, strength r = 0.61 to 0.84, pain r = -0.06, sensation r = -0.43, all p<0.01) [88].

**Motor Activity Log-28.** The Motor Activity Log-28 (MAL-28) is a revision of the MAL-30 with removal of redundant items 'write on paper' and 'put makeup/shaving cream on face' [158]. A single study evaluated the psychometric properties of this measurement tool involving participants greater than 6 months post-stroke, and without any participants with upper limb spasticity [158].

*Content validity*. Content analysis indicated appropriate range of items to cover basic (63%) and instrumental (41%) daily activities in addition to items that require finger movement, bimanual and unimanual tasks [158].

*Results for the whole sample*. Item analysis indicated that 98% of participants encountered included items in daily life [158]. There was evidence for internal consistency (α = 0.94–0.95) and increased test-retest reliability with self-ratings rather than proxy [158]. The MAL-28 held convergent validity with real life measure of hand performance and less with overall physical activity, patient ratings stronger than proxy [158].

**Motricity Index.** The Motricity Index (MI) is a brief scale of motor recovery [55]. Six studies evaluated the psychometric properties of MI [40, 48, 55, 98, 154, 163]; all involved participants post-stroke, and none specifically identified participants with upper limb spasticity.

*Content validity*. Demeurisse, Demol and Robaye [55] detailed the development of the MI with mixed results regarding its relevance and no evidence supporting either comprehensiveness nor comprehensibility.

*Results for whole sample*. There was evidence of the internal consistency of this tool (α = 0.97) [40] and high inter-rater reliability between an experienced and junior doctor

(rho = 0.88) rating 20 participants six weeks post-stroke [48]. The Upper Limb MI (UL MI) correlated strongly with like measures of upper limb activity (RMA arm r = 0.73–0.76) [48] and with global measures of activity (BI r = 0.77) [163] whilst correlating moderately with measures of dexterity (NHPT r = 0.36–0.56) [98]. The UL MI correlated strongly with impairments also, including grip strength (r = 0.74–0.94) [40]. The MI, when combined with the visual neglect recovery index and age at 2–3 days post-stroke was a significant predictor of independence at 3 months (β = 0.042, p < .001) and 6 months (β = 0.038, p < .001) [154]. Evidence of a ceiling effect was noted, with 18% of the sample scoring the maximum score within the UL component of the MI on discharge from a rehabilitation ward post-stroke [98]. There was no evidence of a floor effect.

**Nine-Hole Peg Test.** The Nine-Hole Peg Test (NHPT) is a timed measure of unilateral upper limb dexterity through the placing and removal of nine pegs in/out of a board [178]. Ten studies evaluated the psychometric properties; 5 post-stroke [38, 94, 98, 129] and 5 included participants with MS [39, 51, 79, 139, 150]. One study specifically identified participants with upper limb spasticity [38].

*Content validity*. The NHPT was first discussed as being used in a study in 1985 [179]; no information was reported to inform the development nor content validity of the NHPT.

*Results for whole sample*. The NHPT when used with participants post-stroke correlated with both observed (r = 0.36–0.95) [38, 79, 94, 98, 139] and self-reported measures of activity and hand use (r = 0.53–0.66) [98], was more sensitive than the FAT [94], had poor predictive validity in comparison to like measures, and did not predict HRQoL [129]. The NHPT correlated highly with measures of tremor and dexterity in MS, common activity limitation features (r = -0.62 - -0.87 p<0.005) [180]. There was evidence for the reliability of the NHPT (inter-rater Rho = 0.75–0.99 and test-retest Rho = 0.68–0.90 and 0.83–0.99) when administered to participants 18 months post-stroke [94]. The NHPT was moderate to highly responsive within the first 6 months post-stroke (ES = 0.52–0.66) [38, 98], was more responsive than the upper limb MI [98] and measures of strength, equally responsive to the ARAT, Jebsen-Taylor test of hand function and less responsive than the SIS-hand [38]. True change was indicated by a change of 20% when administered to participants with MS [150]. There were no floor or ceiling effects found in the MS population.

*Results pertaining to sample with upper limb spasticity identified*. Strong correlations with measures of hand use, grip and dexterity were reported in stroke populations ($r_s$ = 0.61–0.95) and with measures of strength ($r_s$ = 0.61–0.82) [38] despite the NHPT being a simulated task performance measure. The NHPT was found to be equally responsive as like measures of upper limb activity performance (ARAT and Jebsen-Taylor test of hand function) (ES 0.52–0.66), more responsive than measures of impairment (pinch and grip strength) but less responsive than the SIS-Hand (ES = 0.55–1.018) in the first 6 months post-stroke [38].

**Oxford Handicap Scale.** The Oxford Handicap Scale (OHS) is a simple tool modified from the Rankin Scale to grade the ability of a person and the level of daily assistance required to live independently [181]. Two studies evaluated the psychometric properties of the OHS, both including participants less than 6 months post-stroke [144, 152]. Neither study specifically identified participants to have upper limb spasticity.

*Content validity*. No published information regarding the development nor content validity of the OHS was located.

*Results for whole sample*. The OHS was not a predictor of caregiver burden [144] but was found to predict both the number of services and amount of time required from services on discharge [152].

**Rivermead Motor Assessment.** The Rivermead Motor Assessment (RMA) [117] is comprised of three sections; for this review studies were separated into two categories 1) 'RMA' all

three sections (upper limb, trunk and leg) administered and reported and 2) 'RMA UL' upper limb section of the RMA only administered and reported. A total of 7 studies were included [25, 26, 48, 117, 129, 147], all studies included participants post-stroke, 4 of the 7 studies included participants less than 6 months post-stroke [26, 48, 101, 147]. When separated into the two categories, evidence for the 'complete RMA' was drawn from 5 studies [25, 26, 101, 117, 147] and evidence for the 'RMA UL' section was drawn from 6 studies [25, 26, 48, 117, 129, 147].

*Content validity*. Test authors Lincoln and Leadbitter [117] detail the measurement tool development. This was completed via selecting a preliminary series of items ranging widely in difficulty ordered into the three sections; gross, leg and trunk and arm. All individual sections were found to have mixed results regarding relevance, reduced due to methods used to create items and nil information regarding comprehensiveness nor comprehensibility.

*Results for whole sample*. The hierarchical scale of the RMA in an acute and non-acute stroke sample found varying results. Evidence to support the scalability of the RMA was found for the gross function and arm section in acute stroke only [26]. Scalability was supported in the gross function section only, when used with participants 6 and 12 months post-stroke [25, 147]. The RMA correlated with ADL performance (r = 0.51) and balance (r = -0.45) [147], a related construct. Agreement between clinician and participants predicted scores with achieved scores was found (clinician ICC 0.965 Bland Altman 96.6; participants ICC 0.908 Bland Altman 79.3) [101]. The hierarchical scale of the RMA UL section was supported only when administered to participants in the acute phase post-stroke (Guttman scaling criteria met) [26], the scalability criteria was not met when used with participants 6 and 12 months post-stroke [25]. The UL section of the RMA was found to correlate strongly with measures of upper limb activity at 6, 12 and 18 weeks post stroke (r = Rho 0.73–0.76) [48] and greater than six months post stroke (r = - 0.80) [129]. The RMA UL correlated moderately with perceived physical activity (r = -0.47) and did not predict overall HRQoL [129].

**Stroke-Adapted Version of the Sickness Impact Profile.** The Stroke-Adapted Version of the Sickness Impact Profile (SA-SIP30) was derived from the original Sickness Impact Profile and contains the following 8 subscales: body care and movement, mobility, ambulation, social interaction, emotional behavior, alertness behavior, communication and household management [160]. Four studies evaluated the psychometric properties of the SA-SIP30 [58, 69, 148, 160], all involved participants post-stroke, and only one study specifically identified participants with upper limb spasticity [58].

*Content validity*. Test authors detailed the methodology applied to create the SA-SIP, based on statistical relevancy and homogeneity [160]. The scale was found to be relevant, however to lack comprehensiveness (as only 5 of 9 recommended dimensions for patient-based, health related quality of life measures were included) [148]. No information regarding comprehensibility was provided.

*Results for whole sample*. The SA-SIP accounted for 53% of variance in predicting participation ($R^2 = 0.63$, P<0.001) and was more sensitive to detecting stroke related changes impacting on independence at 6 months post-stroke [69].

*Results pertaining to sample with upper limb spasticity*. The SA-SIP30 was significantly associated with greater disability in hygiene, dressing, limb posture and pain (P < .05) [58].

**Stroke Impact Scale.** The Stroke Impact Scale (SIS) is a stroke-specific measure of global health outcome [64] and comprises of eight domains: strength, hand function, activities of daily living, instrumental activities of daily living, mobility, communication, emotion, memory and thinking, and participation. The SIS was found to be reported as either individual or collective domains which are administered and reported separately. To maintain consistency across all measures within this review, the SIS was required to be administered in full and in

the form of version 3 to meet inclusion criteria. Ten studies evaluated the psychometric properties of version 3 of the SIS [64–66, 68, 71, 99, 110, 112, 148, 170], all included participants post-stroke and none specifically identified participants with upper limb spasticity.

*Content validity.* The SIS was originally developed following a comprehensive iterative process with the use of participants, caregivers and standardized instrument development guidelines implemented but specific details are not available (unpublished information) [68]. Rasch analysis led to revision of the measure [64] demonstrating comprehensiveness (containing 7 of 9 recommended dimensions for patient-based, health related quality of life) and to be more comprehensive than EQ-5D, SA-SIP and SF-36 [148].

*Results for whole sample.* Rasch analysis refined the SIS into version 3 producing unidimensional domains ranging in item difficulty and with the ability to discriminate [64]. A single index was proposed, aggregated from the 8 domains ($\alpha = 0.93$) accounting for 68.76% of the variance [99]. These 8 domains were each found to be internally consistent ($\alpha \geq 0.86$–0.96) [66, 99], suggesting possible item redundancy and further investigations of shorter forms. Agreement between patient and proxy ratings were fair to excellent, being stronger in the observable physical domains (ICC 0.50 to 0.83) [65]. The tool was reliable between testing sessions when administered via mail (ICC 0.77–0.99) and telephone modes (ICC 0.90–0.99) [66]. The individual and related domains of the SIS were found to correlate with global measures of independence, activity and participation, both patient and proxy reported, (r = 0.69–0.78) [65, 110, 170]. The SIS was able to discriminate between participants deemed recovered by the BI [112] and held superior ability to discriminate between varying levels of disability compared to the FIM and SF-36V (modified version of the SF-36) when tools were administered via phone [110]. Floor and ceiling effects were varied ranging from nil floor effect and 0–32% ceiling effect [71, 110].

**Upper-Limb Motor Assessment Scale.** The Upper Limb -Motor Assessment Scale (UL-MAS) is a subscale of items 6, 7 and 8 of the Motor Assessment Scale, and it provides a task orientated performance-based measure of upper limb activity [46]. Ten studies evaluating the psychometric properties of the UL-MAS were included [46, 100, 103, 109, 116, 118, 119, 126, 137, 146], all involved participants less than 6 months post-stroke, and no studies specifically identified participants with upper limb spasticity.

*Content validity.* Evidence located for the development of the MAS and subsequent UL-MAS did not indicate participants were consulted on the comprehensiveness or comprehensibility of included items [46]. Relevance of items for the intended purpose was sufficient.

*Results for whole sample.* There was evidence to support the production of a single composite score from the UL-MAS items, which may be interpreted as a total score for UL function [116]. Inconsistencies were identified within the hierarchical scoring [126, 137, 146] with clinical recommendations to attempt and score every item [126]. Furthermore, task 2 within the Hand Movements item may not be indicative of upper limb motor recovery in adults aged 65 years and older [126]. The UL-MAS is a unidimensional scale measuring a single construct, upper limb motor performance, ($\alpha = 0.83$ to 0.95, and with removal of wrist deviation 0.93) [100, 116, 126]. It was reliable between (Kendall Tau = 0.74–1.00) and amongst assessors (kappa 0.93–1.0, 88–85% agreement) [46, 118]. The UL-MAS was able to discriminate between differing levels of motor recovery both in the acute and subacute phase, with Rasch based scoring more precise [103]. Varying levels of floor and ceiling effects have been reported for the UL-MAS (floor effect 0–38%, ceiling effect 0–67%) [126, 137, 146].

## Discussion

This systematic review located, appraised and synthesized the body of literature investigating the psychometric properties of measurement tools which assess upper limb function in the

context of everyday activities. Across the included 29 measurement tools, there was wide variability in the quality of evidence in relation to participants with neurological conditions, but overall, tools with the greatest number of psychometric publications demonstrated the strongest evidence. While the FIM™ had the highest quality evidence supporting its validity and reliability, it suffered from both floor and ceiling effects. On consideration of specific constructs measured by the tools, wide variability across quality of evidence remained. Both patient-reported measures, the ArmA and DAS, and performance-based measures, the UL-MAS and ARAT, demonstrated evidence within the measures specifically targeting upper limb activity. Evidence supported use regardless of whether upper limb spasticity was present or not, except for the UL-MAS, which is replaced with the MAL for patients with identified upper limb spasticity. Despite the BI and BI(C&W) holding high to moderate levels of evidence for construct validity, the FIM held the strongest level of evidence for global measures of activity, regardless of whether or not upper limb spasticity was present. The SIS, a patient-reported measure, held the strongest level of evidence across a greater number of properties and demonstrated higher correlations with measures of upper limb performance and activity of the global health-related quality of life measures. The EQ-5D and SA-SIP were the only health-related quality of life measures with evidence supporting construct validity for participants with upper limb spasticity. In light of mixed findings without a clearly superior measurement tool, findings highlights the need for further research into the psychometric properties of measurement tools which capture upper limb activity and/or participation performance.

The search yielded psychometric studies primarily conducted between 2000 and 2010, with an even split of additional evidence located in the 10 years either side of that decade. It was interesting that few papers have been published in the more recent years–this may reflect publication preferences of journals in rehabilitation or a potential assumption by clinicians that the psychometric properties have been well established. Most studies were completed with participants post-stroke in the acute to subacute phase, and as such, findings from these studies may not apply to a more chronic population or a group of neurological clients who have not suffered a stroke. Individual study sample sizes were commonly small (less than n = 100 in over half (56%) of studies), which is a common limitation highlighted by other reviews of functional measurement tools [182, 183]. This finding strengthens earlier calls for continued investment in appropriately powered psychometric studies, inclusion of psychometric evaluation in both routine data collection and longitudinal studies, and a need for scientific journals or outcome tool publishers to publish such research.

The construct validity and responsiveness, followed by reliability properties of measurement tools, were most commonly evaluated across the different tools, but rarely was content validity or measurement error tested. The methodological quality of included studies was wide ranging, from 'inadequate' to 'very good', suggesting that making decisions between measures may be difficult, since there was little consistent data to guide decisions. Detailed data was often lacking within studies such as those reporting on the reliability of tools where information failed to describe testing conditions, stability of patients between sessions and evidence for systematic change occurrence. The COSMIN process recommends that an 'a priori' hypothesis be developed when evaluating construct validity and responsiveness, however in our review only a very small number of studies clearly defined hypotheses about the expected results. The majority of studies were found to report generic hypotheses, where hypotheses were assigned based on interpretations by the authors. Furthermore, the quality of statistical approaches used were low, for example often reporting on statistical significance of findings rather than expected strengths and direction of correlations. Consistent with Zaki and colleagues [184], our review also suggests that the quality of research in psychometrics is unlikely to improve without education and clear guidelines on analysis. The COSMIN checklist may

provide such guidance; the COSMIN process separates the statistical methods based on Classical Test Theory (CTT) or on Item Response Theory (IRT) and an understanding of these methods is likely key to improving the psychometrics of scales where multiple items contribute to an overall score.

The review identified very limited evidence useful for the clinical selection of a single tool to evaluate upper limb activity when upper limb spasticity is present. Inadequate representation of the intended population within the sample of a psychometric study can lead to erroneous assumptions about the psychometrics of a tool [185]. In the context of instrument development, internal and external validity are important for application of an instrument in assessing new target populations (in this case, adults with upper limb spasticity). The DAS, EQ-5D, FIM™, NHPT and SA-SIP had evidence supporting both internal and external validity and responsiveness, however no single measurement tool had identified psychometric evidence for all properties in a sample of participants with upper limb spasticity. This gap in available research is acknowledged, and is both a limitation to this systematic review and a recommendation for further research. The evidence located to guide selection for the broader neurorehabilitation sample was larger in comparison primarily due to additional numbers of contributing studies. However, despite large numbers of contributing studies, we could still not conclude that any of the identified measurement tools from the Ashford and Turner-Stokes [8] review have published psychometric evidence for all relevant psychometric properties.

In this review, despite selecting the most recent and comprehensive set of tools at the time of registering our protocol, we acknowledge a potential limitation in range of tools included and that other existing tools had not been used in clinical trials or cohort studies of patients with spasticity, and therefore were not synthesized in the Ashford and Turner-Stokes [8] review. The limited psychometric testing of the tools that were included was a further limitation, making it difficult to compare the psychometric properties of tools across different pathologies. This may mean that the preferred assessments of a reader does not appear in this extensive review, and where included, it may have only been tested in a single diagnostic population. Only one additional measurement tool beyond the initial systematic review was recommended in the recent national guidelines [13], that tool being the Arm Activity Measure (ArmA). Psychometric studies not published in English were also excluded for pragmatic reasons; formal translations have not yet occurred in many of the measurement tools (e.g. ARAT and UL-MAS) and therefore studies conducted in languages other than English were excluded as per COSMIN guidelines.

## Conclusions

This systematic review provides a comprehensive synthesis of the psychometric properties of the upper extremity measurement tools used to evaluate the dimensions of activity and/or participation. The findings may provide guidance for clinicians on evidence-based measurement tool selection, however further psychometric evaluation of tools is recommended. Together, 29 measurement tools met the inclusion criteria and of these, 8 demonstrated at least a moderate level of confidence in the measurement property estimate in two or more standards. While no tool had at least moderate estimates for all standards (i.e. content validity, structural validity, internal consistency, cross-cultural validity/measurement invariance, reliability, measurement error, criterion validity, hypothesis testing for construct validity and responsiveness), the review was able to suggest which measurement tools should continue to be researched and refined for use. Future research needs to investigate the psychometric properties of these

measurement tools, across a range of neurological populations as well as with a subsample with spasticity in the upper limb.

## Supporting information

**S1 Checklist. PRISMA checklist.**
(DOC)

**S1 File. Search strategy and search terms.** MEDLINE search strategy and terms used in search.
(DOCX)

**S1 Table. Full text exclusion reasons (PRISMA).** This file details reasons for and numbers of studies excluded.
(DOCX)

**S2 Table. Methodological quality and quality criteria ratings.** This file lists all included studies and methodological quality and quality criteria ratings.
(DOCX)

**S3 Table. Summary of results.** This file provides a summary of results for all included studies.
(DOCX)

**S4 Table. Terwee quality criteria and guide for strength of correlations.**
(DOCX)

## Acknowledgments

Many thanks to Jenny Price, Murrumbidgee Local Health District librarian who assisted with sourcing published studies.

## Author Contributions

**Conceptualization:** Shannon Pike, Anne Cusick, Natasha A. Lannin.

**Formal analysis:** Shannon Pike, Anne Cusick, Kylie Wales, Natasha A. Lannin.

**Investigation:** Shannon Pike, Anne Cusick, Kylie Wales, Lisa Cameron, Natasha A. Lannin.

**Methodology:** Shannon Pike, Anne Cusick, Kylie Wales, Lynne Turner-Stokes, Stephen Ashford, Natasha A. Lannin.

**Supervision:** Anne Cusick, Natasha A. Lannin.

**Writing – original draft:** Shannon Pike, Anne Cusick, Natasha A. Lannin.

**Writing – review & editing:** Shannon Pike, Anne Cusick, Kylie Wales, Lisa Cameron, Lynne Turner-Stokes, Stephen Ashford, Natasha A. Lannin.

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
