## [Decision Letter · Decision Letter 0]

23 Oct 2020

Dear Dr. Lannin:Thank you for submitting your manuscript to PLOS ONE. After careful consideration, we feel that it has merit but does not fully meet PLOS ONE's publication criteria as it currently stands. Therefore, we invite you to submit a revised version of the manuscript that addresses the points raised by both reviewers during the review process.Please submit your revised manuscript by November 14th. If you will need more time than this to complete your revisions, please reply to this message or contact the journal office at plosone@plos.org. Please include the following items when submitting your revised manuscript:

We look forward to receiving your revised manuscript.

Kind regards,

Alessandra Solari, M.D.

Academic Editor

PLOS ONE

Journal Requirements:

2.We note that you have indicated that data from this study are available upon request. PLOS only allows data to be available upon request if there are legal or ethical restrictions on sharing data publicly. For information on unacceptable data access restrictions, please see http://journals.plos.org/plosone/s/data-availability#loc-unacceptable-data-access-restrictions.

Reviewers' comments:

Reviewer's Responses to Questions

**Comments to the Author**

1. Is the manuscript technically sound, and do the data support the conclusions?

Reviewer #1: No

Reviewer #2: Yes

2. Has the statistical analysis been performed appropriately and rigorously? 

Reviewer #1: N/A

Reviewer #2: N/A

3. Have the authors made all data underlying the findings in their manuscript fully available?

Reviewer #1: No

Reviewer #2: Yes

4. Is the manuscript presented in an intelligible fashion and written in standard English?

Reviewer #1: Yes

Reviewer #2: Yes

5. Review Comments to the Author

Reviewer #1: This is a systematic review evaluates the quality of measures, and of related psychometric studies, assessing activity and/or participation in adults with upper limb spasticity. A second aim is to explore differences in the quality of the tools for adults with a neurological impairment but without upper limb spasticity.

The subject is interesting and deserves to be investigated. The procedure is well explained and the methodology used to determine the overall quality of the evidence (modified COSMIN GRADE) adequate. There are two major concerns which limit the interpretation of results:

-The second aim stated in the introduction (“differences in psychometric properties for the identified measurement tools for adults with a neurological impairment but without upper limb spasticity will be defined”) is not so clear. Plus, it is hard to reach considering the neurological conditions listed in the inclusion criteria in the Methods, as all these conditions are associated with spasticity (≥ 90% diagnosis of a following neurological condition; Stroke, Multiple Sclerosis, Cerebral Palsy, Traumatic Brain Injury, Anoxia). To define differences in psychometric properties for the identified measurement tools it would be better to include diseases that do not involve spasticity (e.g. Parkinson’s disease or neuromuscular disorders).

-The rationale for inclusion of the 10 metre walk test is not clear. As the authors state, “the 10 metre walk test (10MWT) is a common measure of gait speed. Tools evaluating walking speed are relevant as involuntary and/or impaired arm movements can impact on balance and walking abilities when considering upper limb activity performance.”

Walking speed may correlate with arm movements in neurological diseases involving upper together with lower limbs. As the inclusion criteria did not require spare lower limb function such correlation may be found just because upper limb impairment is associated with lower limb impairment, which does not make the 10 meter test a good measure of upper limb function. The sentence should be modified and the need for including the 10 meter test revised critically and properly justified since the screening phase.

Minor:

-In “Now in 2019, such evidence is available” the year should be updated or removed. It is not clear why the availability in 2019 of a list of measures as in reference [6] (Ashford and Turner-Stokes, 2013) prompted a literature search up to 2016. It is like the study has been started in 2017, so “now in 2019” may be a little confusing.

The meaning of the sentence “a second reviewer screened a random 25% sample against inclusion criteria” should be better clarified as it is not clear what were the purpose and the result of 25% sample screening.

Reviewer #2: The paper reports the results of a survey inquiring psychometric properties of upper limb tools assessing upper limb performances in neurological subjects.

Results highlight moderate confidence level in measurement property in 8 out of 30 selected tools. They also highlight the need of investigating psychometric properties of upper limb assessment tools in neurological subjects with upper limb spasticity.

TITLE

The title is appropriate.

INTRODUCTION

I would add a brief paragraph on the relevant literature summarizing findings from published systematic reviews on upper limb assessment tools.

The aim of this study was “to firstly critically appraise and summarize the quality of the psychometric properties of previously identified upper limb activity “. However authors included tools assessing other domains (e.g. the 10MWT). I agree that the 10MWT can be used as a proxy to assess ‘real-life’ activities however its inclusion in this paper reduces readability. I would recommend to remove these tools from the analyses.

Authors should also provide a rationale for the inclusion of tools mesuring upper limb and activity and participation in real-life activities

Methods.

It is unclear why authors only selected tools published by Ashford and Turner-Stokes. This paper has been published in 2013. A wide range of new tools has potentially been excluded (e.g.Box and Block Test). Authors included the Arm Activity Measure (ArmA) as a newly developed measure. Following this reasoning other tools should be added.

Additionally, I do not understand why inclusion-exclusion criteria list “Undergoing rehabilitation”. I think this has greatly reduced number of collected papers. A clear example is the nine hole peg test. Authors reported less than 10 papers on psychometric properties of this tools that has been extensively studied in MS and other conditions.

Data Analysis

Please add the Terwee’s quality criteria for measurement properties and provide a guide (categorization rules) to quantify the strength of the correlation (“very weak,” 0.20-0.39 “weak,”…).

Results

95% of studies included post stroke participants. I think It is best to remove findings from conditions other than stroke. Likewise a very limited number of papers reported psychometric properties of several tools I think they can be removed from analysis.

In table 2 replace “While sample” with “Whole sample”, column name “Sample” can be misleading.

It would also be useful adding an extra data on Minimal clinically important difference.

Overall I’m concerned about the number of studies included for each tool. For instance, authors reported 5 studies inquiring NHPT psychometric properties for stroke survivors. Similarly they reported only 5 studies including patients with multiple sclerosis. A review published in 2014 reported at least 14 papers investigating psychometric properties of this outcome measure in MS.

Discussion should be better elaborated. Clinical implications should be included guiding clinicians in the choice of the best assessment tool. For example, coming back to the NHPT the most relevant problem in the clinical use of this tool for stroke survivors is the presence of floor effect. This makes NHPT a second line tool to assess upper limb in this condition. The reverse is true for ARAT in MS population.

Comparisons between patient reported outcome and performance scale should also be considered.

Finally, this paper reports findings on scales inquiring upper limb performances and scales on activity and participation the relationship between these two sets of instruments should be discussed. Are there activity and participation scales better representing upper limb impairments in real life activities than others? Are these scales better correlated with scales inquiring upper limb function. Are correlations with these scales better for Patients reported outcomes than functional scales?

6. PLOS authors have the option to publish the peer review history of their article (what does this mean?). If published, this will include your full peer review and any attached files.

Reviewer #1: No

Reviewer #2: No

---

## [Author Response · Author response to Decision Letter 0]

19 Nov 2020

Each reviewer comment has been responded to and addressed - please see the Response to Reviewer letter

---

## [Decision Letter · Decision Letter 1]

17 Dec 2020

PONE-D-20-15560R1

Psychometric properties of measures of upper limb activity performance in adults with and without spasticity undergoing neurorehabilitation – A systematic review.

PLOS ONE

Dear Dr. Lannin,

Thank you for submitting your manuscript to PLOS ONE. After careful consideration, we feel that it has merit but does not fully meet PLOS ONE’s publication criteria as it currently stands. Therefore, we invite you to submit a revised version of the manuscript that addresses the points raised during the review process.

We look forward to receiving your revised manuscript.

Kind regards,

Alessandra Solari, M.D.

Academic Editor

PLOS ONE

Reviewers' comments:

Reviewer's Responses to Questions

**Comments to the Author**

1. If the authors have adequately addressed your comments raised in a previous round of review and you feel that this manuscript is now acceptable for publication, you may indicate that here to bypass the “Comments to the Author” section, enter your conflict of interest statement in the “Confidential to Editor” section, and submit your "Accept" recommendation.

Reviewer #1: (No Response)

Reviewer #2: (No Response)

2. Is the manuscript technically sound, and do the data support the conclusions?

Reviewer #1: Partly

Reviewer #2: Partly

3. Has the statistical analysis been performed appropriately and rigorously? 

Reviewer #1: Yes

Reviewer #2: N/A

4. Have the authors made all data underlying the findings in their manuscript fully available?

Reviewer #1: Yes

Reviewer #2: Yes

5. Is the manuscript presented in an intelligible fashion and written in standard English?

Reviewer #1: Yes

Reviewer #2: Yes

6. Review Comments to the Author

Reviewer #1: All comments have been addressed with the exception of the request on the timed walk test. Not only I had required to modify the sentence but also that the need for including the 10 meter test had to be revised critically and properly justified since the screening phase. I have also noticed that my fellow reviewer asked to address the issue of the 10 meter test. The indicated sentence has been modified, but it seems that no other action was taken in the paper towards the critical revision and proper justification, besides a response to reviewer "Given our sampling frame for selection of assessments to include in this systematic review was drawn from the 2013 published systematic review of measurement tools reported in upper limb studies of spasticity management, and we published our protocol apriori, we respectfully cannot remove 10MWT from our screening".

Reviewer #2: I have read all changes. It seems that authors did not change the manuscript according to reviewers’ suggestions.

The 10MWT has not been removed from the paper. Authors have added this sentence “The 10 metre walk test .... on lower limb activity performance as involuntary and/or impaired arm movements may impact on balance and walking ability”. First, they did not provide references supporting this statement, second disentangling the effect of UL spasticity on gait using TMWT is, at least, challenging.

In addition authors did not removed or added other conditions

I understand the protocol was developed years ago and the importance to comply with the published study protocol.

However these two constrains made the paper less informative:

Psychometric properties of measures in adults with and without spasticity are unclear, and lack of methodological consistence makes decisions between measures difficult.

Additionally, several tools widely used to assesses UL are not reported especially for pathologies different from stroke.

In my opinion these caveats make difficult to “compare the psychometric properties of a wide range of assessment tools and different pathologies”.

7. PLOS authors have the option to publish the peer review history of their article (what does this mean?). If published, this will include your full peer review and any attached files.

Reviewer #1: No

Reviewer #2: No

---

## [Author Response · Author response to Decision Letter 1]

15 Jan 2021

Reviewer 1 Comments 

All comments have been addressed with the exception of the request on the timed walk test. Not only I had required to modify the sentence but also that the need for including the 10 meter test had to be revised critically and properly justified since the screening phase. I have also noticed that my fellow reviewer asked to address the issue of the 10 meter test. The indicated sentence has been modified, but it seems that no other action was taken in the paper towards the critical revision and proper justification, besides a response to reviewer "Given our sampling frame for selection of assessments to include in this systematic review was drawn from the 2013 published systematic review of measurement tools reported in upper limb studies of spasticity management, and we published our protocol apriori, we respectfully cannot remove 10MWT from our screening".

Author’s response

The authorship team apologise the previous revisions did not meet the reviewer requirements- we had added a brief explanation for why we were adhering to original set of tools, but now see that the reviewers both prefer not to include a LL measure. Therefore, we have now removed the 10MWT from the included outcome measurement tools, with reasoning for this exclusion stated in the Method section.

Reviewer 2 Comments 

I have read all changes. It seems that authors did not change the manuscript according to reviewers’ suggestions.

The 10MWT has not been removed from the paper. Authors have added this sentence “The 10 metre walk test .... on lower limb activity performance as involuntary and/or impaired arm movements may impact on balance and walking ability”. First, they did not provide references supporting this statement, second disentangling the effect of UL spasticity on gait using TMWT is, at least, challenging.

In addition authors did not removed or added other conditions

I understand the protocol was developed years ago and the importance to comply with the published study protocol.

However these two constrains made the paper less informative:

Psychometric properties of measures in adults with and without spasticity are unclear, and lack of methodological consistence makes decisions between measures difficult.

Additionally, several tools widely used to assesses UL are not reported especially for pathologies different from stroke.

In my opinion these caveats make difficult to “compare the psychometric properties of a wide range of assessment tools and different pathologies”. 

Author response:

We have now removed the 10MWT from the included outcome measurement tools with reasoning for this exclusion stated in the Method section. 

Thank you for understanding the importance of our compliance with the registered and published study protocol. We do wish to correct that, as outlined in our paper, the diagnoses included in our systematic review were neurological conditions (not limited only to “stroke”). The search terms are outlined in Table 1, and these included Stroke, Multiple Sclerosis, Cerebral Palsy, Traumatic Brain Injury, and/or Anoxia (hypoxia). 

We appreciate that there are tools used outside of these populations not reported and that there are tools which have not been psychometrically studied in a neurological population such that the tools have thus not been able to be included. We did, however, seek to include any additional tools recommended in clinical practice guidelines (in addition to the tools identified in the Ashford review). These points are outlined in our method.

We additionally appreciate that it may have been the reviewer’s point that without source studies, it is difficult to meet our study aim. Therefore, we have now also added this point to the limitations discussion of our paper: 

“The limited psychometric testing of the tools that were included was a further limitation, making it difficult to compare the psychometric properties of tools across different pathologies. This may mean that the preferred assessments of a reader does not appear in this extensive review, and where included, it may have only been tested in a single diagnostic population.”

---

## [Editor Report · Decision Letter 2]

18 Jan 2021

Psychometric properties of measures of upper limb activity performance in adults with and without spasticity undergoing neurorehabilitation – A systematic review.

PONE-D-20-15560R2

Dear Dr. Lannin,

We’re pleased to inform you that your manuscript has been judged scientifically suitable for publication and will be formally accepted for publication once it meets all outstanding technical requirements.

Kind regards,

Alessandra Solari, M.D.

Academic Editor

PLOS ONE

---

## [Editor Report · Acceptance letter]

27 Jan 2021

PONE-D-20-15560R2 

Psychometric properties of measures of upper limb activity performance in adults with and without spasticity undergoing neurorehabilitation – A systematic review. 

Dear Dr. Lannin:

I'm pleased to inform you that your manuscript has been deemed suitable for publication in PLOS ONE. Congratulations! Your manuscript is now with our production department. 

Kind regards, 

on behalf of

Dr. Alessandra Solari 

Academic Editor

PLOS ONE